# Altered B cell activation contributes to the immunopathogenesis of childhood arthritis-associated uveitis

Bethany R. Jebson[1,2,3], Benjamin Ingledow[1,3], Vicky Alexiou[1,3], Jakub Kubiak[4], Persephone Jenkins[1,3], Yuxuan Meng[4], Melissa Kartawinata[1,2], Restuadi Restuadi[1,2], Wei-Yu Lin[5,6], Chris Wallace[5,6], Colin J. Chu[4,7], Ameenat Lola Solebo[2,8], Lucy R. Wedderburn[1,2,8,9] & Elizabeth C. Rosser[1,3] ✉
On behalf of the CLUSTER consortium*

In Juvenile Idiopathic Arthritis (JIA), the most common childhood rheumatic disease, many patients also develop uveitis (JIA-uveitis), risking life-long vision loss. The mechanisms driving uveitis development in JIA remain understudied. Here, we demonstrate that peripheral blood CD19⁺IgD⁻CD27⁻ double negative type 1 (DN1) B cells are elevated in JIA-uveitis compared to JIA patients without eye disease (JIA). The B cell receptor (BCR) repertoire was also more clonal and somatically hypermutated in JIA-uveitis and antigen-activated B cells infiltrated chronically inflamed JIA-uveitis eyes. Features of heightened B cell activation were recapitulated in experimental autoimmune uveoretinitis (EAU) and disrupting B and T cell interactions using monoclonal antibodies and transgenic mice suppresses uveitis. Together, these findings support a conceptual shift that uveitis is a primarily T cell driven disease and provide evidence for potential new therapeutic strategies that also consider B cells as drivers in disease pathology.

Juvenile Idiopathic Arthritis (JIA) is an umbrella term encompassing a group of arthritides that develop in children under 16 years old. In the most common forms of the disease (oligo-articular and rheumatoid factor (RF)ⁿᵉᵍpoly-articular JIA) up to 30% of patients can develop chronic anterior uveitis[1]. The connection between eye inflammation (uveitis) and joint inflammation (arthritis) in JIA remains unclear[2]. Response to therapy is heterogeneous, with some children developing a treatment-refractory form of disease, leading to life-long sight-loss[2]. Up to 30% of JIA-uveitis patients have lost vision in at least one eye by the age of 18[3]. Rapid control of inflammation is crucial for preventing uveitis-associated visual disability[1–3]. However, therapeutics are still

applied in a stepwise approach. Currently, methotrexate is the first-line therapy for children whose disease remains uncontrolled following treatment with topical steroids, while biologics (e.g., Tumour Necrosis Factor alpha - TNFα - blockade) are given to children whose disease is resistant to both topical steroids and methotrexate[2]. It is estimated that over 25% of children will not respond adequately to TNFα therapy, leading to permanent sight-loss[3]. New studies are needed to understand why uveitis develops in only some JIA patients and whether newly uncovered mechanisms can be used to identify novel therapeutic strategies aiming to improve long-term outcomes in more patients.

[1]Centre for Adolescent Rheumatology at UCL, UCLH and GOSH, London, UK. [2]UCL GOS Institute of Child Health, London, UK. [3]Division of Medicine, UCL, London, UK. [4]UCL Institute of Ophthalmology, London, UK. [5]MRC Biostatistics Unit, University of Cambridge, Cambridge, UK. [6]Cambridge Institute of Therapeutic Immunology and Infectious Disease (CITIID), Jeffrey Cheah Biomedical Centre, University of Cambridge, Cambridge, UK. [7]NIHR Biomedical Research Centre at Moorfields Eye Hospital NHS Foundation Trust and UCL Institute of Ophthalmology, London, UK. [8]Great Ormond Street Hospital for Children NHS Trust, London, UK. [9]NIHR Biomedical Research Centre at Great Ormond Street Hospital for Children NHS Trust, London, UK.*A list of authors and their affiliations appears at the end of the paper. ✉e-mail: e.rosser@ucl.ac.uk

Although current therapeutic strategies for JIA-uveitis are based upon treatments which block inflammatory mediators such as TNFα and most experimental medicine studies have focused on the role that T cells play in uveitis[4,5], there is evidence that B cells may also be important in disease pathogenesis. The primary risk factors for uveitis development in JIA patients is anti-nuclear autoantibody (ANA) positivity[6], a hallmark for a breakdown in B cell tolerance[7], and early age of arthritis onset, which is associated with a prominent B cell transcriptional signature within total peripheral blood mononuclear cells (PBMC) when compared to children with later disease-onset[8]. In patients with JIA-uveitis who develop complications associated with treatment non-response and vision loss requiring surgical intervention, ocular samples provide preliminary evidence that B cells may infiltrate the ocular compartment. Antibody-producing plasma cells can be found within the inflamed iris of ANA+ patients with childhood-onset uveitis, including those with a JIA diagnosis[9,10] and there is intraocular upregulation of B cell-encoded gene expression (e.g., Marginal Zone B and B1 specific protein – MZB1) and B cell-activating and survival factors (e.g., B cell activating factor – BAFF, A proliferation inducing ligand – APRIL) in the aqueous humour of JIA-uveitis patients when compared to uveitis-free patients[11]. Recent experimental uveitis studies in mice have also demonstrated that depletion of B cells using monoclonal antibodies suppresses disease severity[12].

In recent years, there has been a growing interest in understanding how double-negative B cell subsets, or DN B cells, are linked to autoimmune pathogenesis. In healthy adults, DN B cells account for only about 5% of the total B cell population, making them relatively rare[13–15]. Broadly characterised by the absence of CD27 and IgD expression, these cells represent a heterogeneous population with four distinct subtypes. Briefly, DN1 B cells, which constitute the majority of DN B cells in healthy individuals, are characterised as CD11c$^-$CXCR5$^+$CD21$^{+16,17}$. Data has described this subset both as a precursor of switched memory B cells and recent germinal center (GC) emigrants[16] and as a novel durable subset of memory B cells potentially derived following extrafollicular (EF) activation of B cells[18]. Recent studies have linked an expansion of CD27$^-$CD21$^+$DN1-like B cells to dysregulated GC responses in adult conditions such as IgA nephropathy[19], and a trend for increased DN1/DN2 ratio has been described as a feature of defective GC selection in Primary Antiphospholipid Syndrome[20]. In contrast, DN2 B cells (CD11c$^+$CXCR5$^-$CD21$^-$Tbet$^+$) are a less common subset that are suspected to solely originate from the EF response and serve as precursors of plasma cells. In systemic lupus erythematosus (SLE), DN2 B cells are expanded, and the presence of these cells has been correlated with disease severity[13]. Remaining minority subsets of DN B cells include CD11c$^-$CD21$^-$DN3 B cells, with trajectory analysis suggesting that they may serve as precursors of DN2 cells, and DN4 cells, which serve as precursors of IgE class-switched memory B cells in allergic situations[21]. In the context of JIA, DN2 B cells have been shown to accumulate in the inflamed joints of ANA+ patients[22], but no studies have explored the phenotype and function of the breadth of DN B cell subsets across multiple immune cell compartments in JIA or JIA-uveitis.

Unlike other autoimmune conditions, the anti-TNFα Adalimumab remains the only NICE-approved biologic for JIA-uveitis treatment[3]. An unmet need for evidence-based studies that aim to uncover specific uveitic-driving mechanisms which can be exploited for therapeutic innovation remains. To address this gap, we stratified a large cohort of JIA patients recruited to the CLUSTER JIA consortium[23] based on their uveitis status and irrespective of international league of associations for rheumatology (ILAR) subtype[24]. In this cohort, we found that JIA-uveitis patients can be distinguished from JIA patients with arthritis alone (JIA) by a significant expansion of CD19$^+$IgD$^-$CD27$^-$ DN B cells, and specifically DN1 B cells, in the peripheral blood. We found that this increase in DN1 B cells was associated with an increase in developmentally-linked memory B cells and that the B cell receptor (BCR) repertoire showed higher clonality and increased levels of somatic hypermutation in JIA-uveitis compared to JIA. Using a resource of ocular samples from patients with treatment-resistant, severe JIA-uveitis, we found that antigen-activated B cell subsets, and particularly plasmablasts, can be found infiltrating the eye of JIA-uveitis patients. In the mouse model of experimental autoimmune uveoretinitis (EAU), we found evidence of heightened B cell activation, including an increase in GC B cells and plasmablasts in the spleen of mice with EAU compared to naïve controls and an infiltration of B cells into the inflamed ocular compartment of mice with severe EAU. Disrupting B and T cell interactions using antagonistic anti-CD40L monoclonal antibodies[25,26] and BCL6$^{fl/fl}$CD4$^{cre}$ mice, which are deficient in T follicular helper (Tfh) cells[27], we found that uveitis incidence and severity was dramatically reduced in both conditions. Importantly, this study provides the first evidence that anti-CD40L antagonism, which shows efficacy in clinical trials for other B cell mediated autoimmune conditions[25,28], may be beneficial for the treatment of JIA-uveitis and potentially other forms of childhood and adult-onset uveitis.

## Results

### DN B cells, and particular DN1 B cells, are expanded in the peripheral blood of JIA-uveitis patients compared to JIA patients with no eye disease

To address whether the B cell compartment of JIA-uveitis patients is altered compared to JIA patients who do not develop uveitis, a large cohort of JIA ($n = 158$) patients recruited to the CLUSTER consortium were stratified based on their uveitis status. 'JIA' samples ($n = 116$) had no history of uveitis, whilst 44 'JIA-uveitis' samples were categorised by either having previous uveitis or active eye inflammation at the time of sample (Table 1). Assessment of the differential phenotype of 34 different immune cell populations (Supplementary Table S1) in the peripheral blood of JIA-uveitis and JIA patients demonstrated that there was an expanded population of CD19$^+$CD27$^-$IgD$^-$ DN B cells, and particularly CD11c$^-$ DN B cells, in JIA-uveitis compared to JIA patients (Fig. 1A–F and Supplementary Figs. 1–4). No other immune subset was found to be significantly altered between these patient groups after corrections for multiple testing. These differences were specific to the peripheral blood as no differences in immunophenotype could be found when comparing synovial fluid mononuclear cells from JIA-uveitis versus JIA patients (Supplementary Fig. 5). To determine whether these observed increases in DN and CD11c$^-$ DN B cells were being driven by uveitis activity, we stratified our JIA-uveitis patients into those with active and inactive eye inflammation at the time of sample. The increases in DN and CD11c$^-$ DN B cells were observed in JIA-uveitis patients regardless of uveitis disease activity (Fig. 1G, H). In a small cohort of JIA patients who went on to develop uveitis after the sample date, we also saw that this increase in DN B cells did not precede uveitis development (Supplementary Fig. 6).

It should be noted that this JIA-uveitis cohort is enriched for ANA-positive oligoarticular JIA patients, reflecting the established epidemiological pattern where this subgroup of JIA carries the highest risk for uveitis development[29]. Thus, we next performed a subgroup analysis restricted to oligoarticular patients only to ensure our findings were not driven by subtype heterogeneity (Fig. 1I, J). This analysis replicated the findings of increased CD19$^+$CD27$^-$IgD$^-$ DN and CD11c$^-$ DN B cells in JIA-uveitis patients, though with reduced statistical power due to smaller sample sizes (DN: $p = 0.0161$, DN1: $p = 0.0083$). To exclude that the increase in CD11c$^-$ DN B cells in peripheral blood was due to the impact of various clinical factors further to an oligoarticular subtype that increase the risk of developing JIA-uveitis, such as a positive ANA titre or a young age of

**Table 1 | Patient demographics. Characteristics of JIA-uveitis and JIA patients included within the study**

| Characteristic | JIA-uveitis | % Missing data | JIA | % Missing data | P-value ( ≤ 0.05 values shown) |
|---|---|---|---|---|---|
| **Total samples (n)** | 44 | – | 116 | – | < 0.0001 |
| **Total patients (n)** | 43 | | 115 | | < 0.0001 |
| Active uveitis n (%) | 29 (65) | | N/A | | N/A |
| Recruitment years | 2010– 2019 | | 1999–2019 | | N/A |
| Age at sample, years, median, (range) | 8.7 (1.7 – 16) | 0 | 9.1 (1.2 – 16.6) | 0 | – |
| Age at disease onset, years, median (range) | 4.3 (0.4 – 11.9) | 0 | 5.6 (0.2 – 16) | 2 | – |
| **Sex n (%)** | | | | | |
| Female | 35 (79) | 0 | 74 (64) | 0 | – |
| Male | 9 (21) | 0 | 42 (36) | 0 | – |
| **Ancestry n (%)** | | | | | |
| Non-Caucasian | 15 (35) | 0 | 25 (22) | 0 | – |
| Caucasian | 29 (65) | 0 | 91 (78) | 0 | – |
| **JIA subtype n (%)** | | | | | |
| Oligoarticular – persistent | 15 (34) | 0 | 12 (10) | 0 | 0.0007 |
| Oligoarticular – extended | 16 (36) | 0 | 26 (22) | 0 | – |
| Polyarticular RF-ve | 11 (25) | 0 | 44 (38) | 0 | 0.0008 |
| Polyarticular RF+ve | 2 (5) | 0 | 6 (5) | 0 | – |
| Enthesitis-related | 0 (0) | 0 | 18 (16) | 0 | 0.0036 |
| Psoriatic JIA | 0 (0) | 0 | 7 (6) | 0 | – |
| Undifferentiated | 0 (0) | 0 | 1 (1) | 0 | |
| 'Polygo' | 42 (96) | 0 | 82 (70) | | 0.0005 |
| Active joint count, median, (range) | 4 (0-23) | 2 | 4 (0 – 34) | 0 | – |
| ANA positive n (%) | 37 (84) | 0 | 61 (59) | 3 | 0.0074 |
| RF positive n (%) | 4 (10) | 7 | 11 (10) | 9 | – |
| HLA-B27 positive n (%) | 1 (8) | 70 | 16 (30) | 54 | |
| **Treatment at time of sample n (%)** | | | | | |
| MTX | 16 (36) | 0 | 18 (15) | 0 | 0.0084 |
| Anti-TNF | 2 (5) | 0 | 0 (0) | 1 | – |
| Anti-IL6 | 0 (0) | 0 | 0 (0) | 1 | – |
| Topical eye steroids | 14 (32) | 0 | 0 (0) | 0 | N/A |
| Systemic steroids | 8 (18) | 0 | 5 (4) | 1 | 0.0081 |

'Polygo' = combined oligoarticular and polyarticular RF- JIA subtypes, RF = Rheumatoid Factor, ANA = Anti-nuclear antibody, HLA-B27 = Human Leucocyte Antigen B27, MTX = Methotrexate. Statistical significance was determined using a two-tailed Chi-squared test or two-tailed Fisher's exact test (used when any expected cell count was < 5). Exact p-values are shown for comparisons with p ≤0.05; variables not applicable to both groups are marked N/A.

arthritis onset or presenting with polyarticular-RF-negative as well as oligoarticular arthritis subtype (together sometimes referred to as 'polygo' types of JIA) which are known to impact B cell phenotype[30,31], a multiple linear regression analysis was performed. As expected, variables such as ethnicity, 'polygo' subtype, methotrexate treatment and ANA status had a significant association with the CD11c⁻ DN B cell expansion. However, this analysis demonstrated that the strongest driving factor behind the observed increase in CD11c⁻ DN B was a positive uveitis status (Fig. 1K).

DN B cells are a heterogeneous group of cells which can be generated through different B cell activation pathways with described DN B cell subsets including CD11c⁻CXCR5⁺DN1 B cells, CD11c⁺CXCR5⁻DN2 B cells, CD11c⁻CXCR5⁻ DN3 B cells, and a minority subset of IgE+ CXCR5⁺DN4 B cells[17]. Thus, an in-depth exploration of the phenotype of CD11c- DN B cells was performed on a subset of our original full JIA and JIA-uveitis patient cohort. The expanded cells were CXCR5⁺CD11c⁻, confirming that DN1 B cells, and no other DN B cell subset, were expanded in JIA-uveitis compared to JIA (Fig. 2A–D). The expanded DN1 B cells in JIA-uveitis peripheral blood were also CD86⁺, a key activation marker involved in B:T cell interactions[32], and CXCR3⁺, which controls migration of B cells into the inflamed site and the nervous system[33] (Fig. 2E–H).

**The expansion in DN1 B cells in JIA-uveitis is associated with an increase of developmentally linked memory B cells and a more clonal B cell repertoire**

DN1 B cells have been previously postulated to readily differentiate into memory B cells[17]. Accordingly, in the full cohort of JIA and JIA-uveitis patients we found a highly significant and strong positive correlation between the proportions of CD11c⁻DN1-like B cells and CD24ʰⁱCD38⁻ memory B cells (Fig. 3A). In a penalised regression 'LASSO' regression model where all the phenotypic flow cytometry data from all enumerated 34 immune cell populations were inputted alongside all the recorded clinical demographic information for each patient, the model also deemed that factors most influencing the likelihood of a JIA patient having uveitis were the increase in CD11c⁻DN1-like B cells alongside CD24ʰⁱCD38⁻ memory B cells followed by decreased in CCR6⁻CXCR3⁻ Th2 T cells, a positive ANA status and an oligoarticular JIA subtype (Fig. 3B). Interestingly, when comparing model accuracy, the variables identified by the LASSO regression (area under curve, AUC 0.82) outperformed both a multiple logistic regression model incorporating key clinical risk factors (JIA subtype, ANA status, age, and sex) (AUC 0.74) and a model using ANA as the sole predictor (AUC 0.64) (Fig. 3C). Previous studies interrogating the developmental connection between DN1 B cells and memory B cells have been performed in adults and recent studies have suggested that

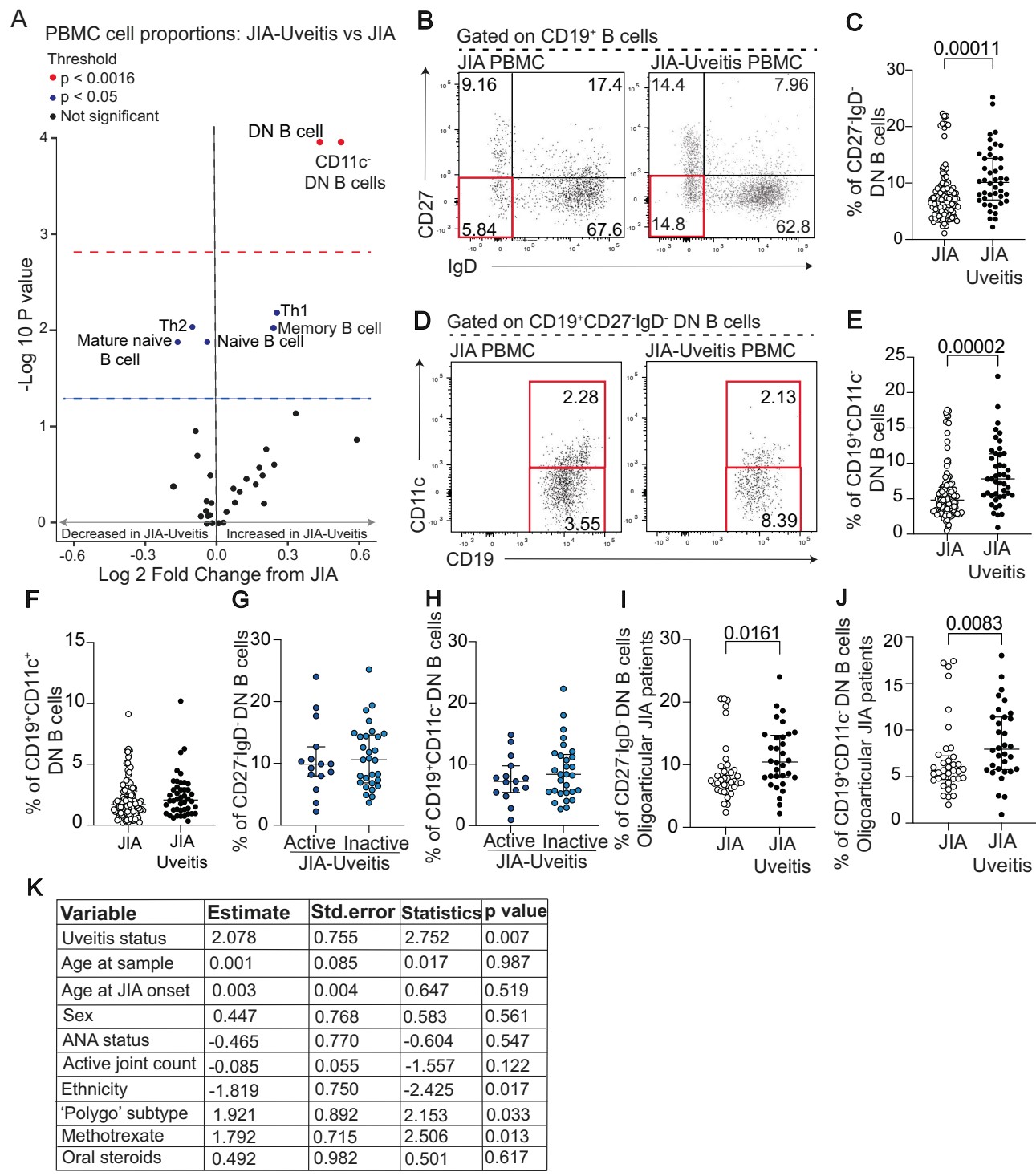

**K**

| Variable | Estimate | Std.error | Statistics | p value |
|---|---|---|---|---|
| Uveitis status | 2.078 | 0.755 | 2.752 | 0.007 |
| Age at sample | 0.001 | 0.085 | 0.017 | 0.987 |
| Age at JIA onset | 0.003 | 0.004 | 0.647 | 0.519 |
| Sex | 0.447 | 0.768 | 0.583 | 0.561 |
| ANA status | -0.465 | 0.770 | -0.604 | 0.547 |
| Active joint count | -0.085 | 0.055 | -1.557 | 0.122 |
| Ethnicity | -1.819 | 0.750 | -2.425 | 0.017 |
| 'Polygo' subtype | 1.921 | 0.892 | 2.153 | 0.033 |
| Methotrexate | 1.792 | 0.715 | 2.506 | 0.013 |
| Oral steroids | 0.492 | 0.982 | 0.501 | 0.617 |

DN1 B cells may also act as a long-lived memory population derived via EF activation pathways[18]. Thus, to interrogate B cell developmental pathways in children and specifically JIA patients, we next performed trajectory analysis on the integrated CD11c- B cell compartment from the peripheral blood of a subset of JIA and JIA-uveitis patients. We delineated 9 clusters of peripheral blood B cells, including transitional B cells, naïve B cells, CXCR3- and CXCR3+ unswitched memory B cells, CXCR3- and CXCR3+class-switched B cells, DN1 B cells, plasmablasts and plasma cells (Fig. 3D, E). Trajectory analysis inferred that there were three lineages by which B cells could differentiate using these data and included two pathways by which DN1 B cells could further differentiate. In lineage 1, DN1 B cells differentiated into class-switched memory B cells, which

represented the end of the differentiation pathway. In lineage 2, DN1 B cell differentiated into plasma cells/plasmablasts with a minority population of class-switched memory B cells acting as an intermediate step. A third pathway (lineage 3) excluded DN1 B cells and was the provenance of a population of potentially long-lived unswitched memory B cells (Fig. 3E).

To next address whether the expansion in B cell subsets had impacted both B cell receptor repertoire and transcriptional profile of B cells in JIA-uveitis compared to JIA patients, bulk CD19+ RNAseq data from JIA ($n = 101$) and JIA-uveitis patients ($n = 33$) from the CLUSTER cohort was used to perform both BCR repertoire analysis and differential gene expression analysis[34]. Although there were limited differences in the total B cell transcriptome when comparing JIA and JIA-

**Fig. 1 | CD11c⁻ DN B cells are significantly expanded in the peripheral blood of JIA-uveitis patients compared to JIA patients.** All data were generated from PBMC collected from JIA patients with no uveitis (JIA, *n* = 116) and JIA patients with uveitis (JIA-uveitis, *n* = 44) unless otherwise stated. **A** Volcano plot showing all 34 measured cell populations within the PBMC of JIA-uveitis patients. *P*-values were calculated using two-tailed Mann-Whitney U tests comparing JIA-uveitis (*n* = 44) and JIA (*n* = 116) groups for each cell population. Significance thresholds were set at *p* ≤0.05 (blue dashed line) and *p* ≤0.0015 (red dashed line; Bonferroni-corrected threshold, 0.05/34 populations). **B** Representative flow cytometry plots showing the frequency of (**C**). CD27⁻IgD⁻ Double Negative (DN) B cells. **D** Representative flow cytometry plots and dot plots showing the frequency of (**E**) CD27⁻IgD⁻CD11c⁻ DN B cells and (**F**) CD27⁻IgD⁻CD11c⁺ DN B cells within CD19⁺ live singlets. Dot plots showing the frequency of (**G**) CD27⁻IgD⁻DN B cells and (**H**) CD27⁻IgD⁻CD11c⁻ DN B cells within CD19⁺ live singlets from JIA patients with active and inactive uveitis (active JIA-uveitis, *n* = 15, navy blue symbols) and (inactive JIA-uveitis, *n* = 29, light blue symbols). Dot plots showing the frequency of (**I**) CD27⁻IgD⁻ DN B cells and (**J**). CD27⁻IgD⁻CD11c⁻ DN B cells within CD19⁺ live singlets from patients with oligoarticular arthritis only (JIA, *n* = 37) and (JIA-uveitis, *n* = 30). Significance of difference between groups was determined using two-tailed Mann-Whitney tests. *P*-values below or equal to 0.05 are shown on graphs, but the significance threshold was set at ≤ 0.005 to adjust for multiple testing for phenotypic analysis (**B–J**). Error bars represent median ± IQR for groups. **K** Table shows results of a multiple linear regression model on the impact of predictor variables (uveitis status, age at sample, age at JIA onset, sex, ANA status, active joint count, ethnicity and 'Polygo' subtype, methotrexate treatment and oral systemic steroid treatment) on the dependant variable, CD11c⁻ DN B cells. The association between CD11c⁻ DN B cell frequency and clinical variables was assessed using multiple linear regression. *P*-values for individual coefficients were calculated using two-sided *t* tests. Uveitis status (*p* = 0.007), ethnicity (*p* = 0.017), 'polygo' subtype (*p* = 0.033), and methotrexate treatment (*p* = 0.013) were significantly associated with CD11c⁻ DN B cell frequency.

uveitis patients (Supplementary Fig. 7), there were differences in the BCR repertoire. More specifically, when adjusted for age and sex, the JIA-uveitis group exhibited a reduced diversity score compared to JIA, suggesting a more clonal B cell receptor repertoire (Fig. 3F). When assessing BCR mutational load, which reflects somatic hypermutation levels, we also found that JIA-uveitis patients exhibited higher mean mutational frequency than JIA patients (Fig. 3G). Of note, these changes in BCR diversity were not accompanied by differences in the length of the CDR3 amino acid sequence (Supplementary Fig. 7C) or in the utilisation of functional V genes (Supplementary Fig. 7D). However, there was a significant enrichment in the V pseudogene IGHV3.60 compared to JIA alone (Supplementary Fig. 7D).

### B cells can be found in the ocular compartment of JIA-uveitis patients and are mainly of a plasma cell phenotype

Despite our observed changes in the B cell compartment between JIA and JIA-uveitis patients in the periphery, to fully elucidate any potential disease mechanisms it is necessary to interrogate the active disease site – the eye. To address this, we analysed rare aqueous humour (AqH) samples collected from JIA-uveitis patients (*n* = 2) undergoing cataract surgery using spectral flow cytometry (Supplementary Table 5). This demonstrated that leucocyte proportions were similar between these patients (Fig. 4A). Using unsupervised clustering, we were able to identify six clear immune cell populations within the AqH-infiltrating leucocytes including CD4⁺ T cells, CD8⁺ T cells, innate-like lymphocytes, granulocytes, monocytes and a minority subset of CD19⁺ B cells in all patients (Fig. 4B). Although the minority subset, further characterisation of the phenotype of AqH-infiltrating B cells demonstrated that B cells present within the AqH were mainly of a class-switch memory (CD19⁺CD27⁺IgD⁻CD38⁻CD20⁺) or plasmablast (CD19⁺CD27⁺IgD⁻CD38⁺CD20⁺) phenotype (Fig. 4C, D). It has been previously shown that whilst there is a limited number of CD19⁺ B cells in the synovial fluid of JIA patients, that there is a larger B cell infiltrate of mainly plasma cells into the synovial tissue itself[35–37]. To address if this was also the case in JIA-uveitis, we analysed archival H&E stained tissue from enucleated eyes collected from JIA-uveitis patients (Supplementary Table 6). In all samples (*n* = 3, patient 1, 2 & 3), there was clear evidence of plasma cell infiltration based on their classic morphology, including clockface nuclei and large cytoplasmic domains (Fig. 4E–G). Infiltration of plasma cells varied in location depending on the sample, but evidence of plasma cell infiltration was found in the iris of patients 1 & 2, the cornea of patient 2 and the choroid of patient 3 (Fig. 4E–G).

### Disruption of B and T cell interactions suppresses experimental autoimmune uveitis severity

Although our human studies suggested that B-cell activation pathways are dysregulated in JIA-uveitis patients, they do not address whether these pathways are directly contributing to uveitis pathogenesis. Thus, we next sought to understand whether the animal model of experimental autoimmune uveoretinitis (EAU) could be used to perform mechanistic studies to understand the direct contribution of B cells to uveitis pathology. Although EAU does not fully replicate JIA-uveitis as it lacks joint inflammation (arthritis), it does serve as a valuable tool to model the breakdown of the blood-retina barrier, which occurs in human uveitis[38]. As EAU is primarily characterised as a T cell driven disease[4], we first assessed how the peripheral and ocular B cell immunophenotype was altered in mice with EAU compared to controls. This showed that there a significant increase in multiple splenic B cell subsets in mice with EAU versus controls including CD95⁺GL7⁺ germinal centre (GC) B cells and CD138⁺Blimp-1⁺ plasmablasts (Fig. 5A–D). Notably, the frequency of PD1⁺CXCR5⁺ Tfh cells was also increased in the spleens of EAU mice compared to controls (Fig. 5E, F). We also observed that there was a significant increase in CD19⁺ B cells in the ocular compartment of mice with EAU compared to controls, with a specific infiltration of B cells into the ocular compartment of mice with severe disease (Fig. 6A–D). We next assessed the efficacy of anti-CD40L, which has been previously used to block T cell-dependent B cell activation[39], in modulating EAU disease severity. This demonstrated that mice treated with antagonistic anti-CD40L were resistant to EAU induction and had a significantly reduced frequency of GC B cells and Tfh T cells when compared to isotype control treated mice (Fig. 7A–G). Secondly, we assessed the severity of in BCL6^fl/fl^CD4^cre^ mice, which are deficient in Tfh cells, leading to impaired B cell responses[27]. The severity of EAU was also significantly reduced in BCL6^fl/fl^CD4^cre^ mice compared to control mice, and there was a reduction in the frequency of GC B cells and Tfh T cells (Fig. 8A–G).

## Discussion

The development of uveitis can be a severe complication following a JIA diagnosis, and this common co-morbidity can lead to lifelong visual disability in children already struggling with symptoms of arthritis. Despite this, there are comparatively few mechanistic studies investigating the processes driving ocular inflammation compared to joint inflammation in JIA patients. In this study, we show that there is dysregulated B cell activation in JIA-uveitis compared to JIA patients with arthritis alone. We also identify that disrupting B-T cell interactions, namely through anti-CD40L antagonism, may be a new treatment target for JIA-uveitis, which may be efficacious for those children that do not respond to first line therapies.

In JIA-uveitis, several clinical factors increase the likelihood of a JIA patient developing eye disease[31]. This was reflected in our cohort as JIA patients who were younger, female, ANA positive and/or had an oligoarticular subtype of JIA were over-represented within the JIA-uveitis patient subgroup. As these multiple clinical factors could confound results by influencing the frequency of certain B cell populations in JIA and JIA-uveitis patients independently of uveitis studies[8,40], we used both supervised and unsupervised regression analyses to control for

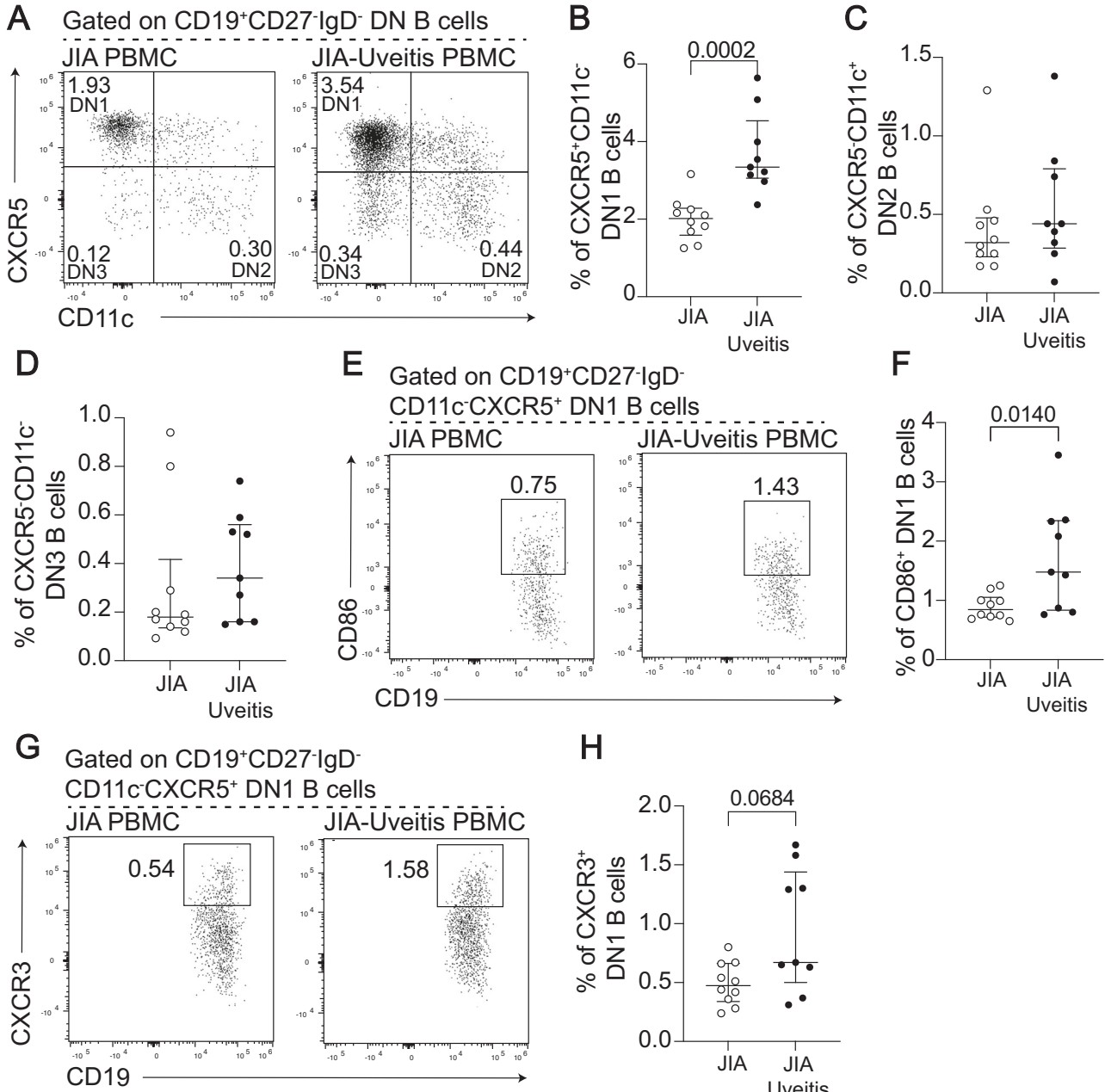

**Fig. 2 | DN1 B cells are significantly expanded in the peripheral blood of JIA-uveitis patients compared to JIA patients and show features of altered activation and migration.** All data were generated from PBMC collected from JIA patients with no uveitis (JIA, *n* = 10) and JIA patients with uveitis (JIA-uveitis, *n* = 9). **A** Representative flow cytometry plots and dot plots showing the frequency of (**B**) CXCR5$^+$CD11c$^-$ DN1 B cells, (**C**) CXCR5$^-$CD11c$^+$ DN2 B cells and (**D**) CXCR5$^-$CD11c$^-$ DN3 B cells within CD27$^-$IgD$^-$ DN B cells. Representative flow cytometry plots (**E**) and dot plots (**F**) showing the frequency of CD86$^+$ DN1 B cells within CD19$^+$ B cells. Representative flow cytometry plots (**G**) and dot plots (**H**) showing the frequency of CXCR3$^+$ DN1 B cells within CD19$^+$ B cells. Significance of difference between groups was determined using the two-tailed Mann-Whitney test. *P*-values below or equal to 0.05 are shown on graphs, but the significance threshold was set at ≤ 0.005 to adjust for multiple testing (Bonferroni correction). Error bars represent median ± IQR for groups.

potential confounders. Firstly, a subgroup analysis on the more homogenous group of oligoarticular JIA patients alone demonstrated that we still observed an increase in DN and DN1 B cells in JIA patients with uveitis compared to those with JIA alone. Secondly, in a linear regression model, uveitis status had the most significant effect on the expansion of CD11c$^-$DN1-like B cells, even when accounting for potential confounders. Importantly, when we stratified JIA-uveitis patients by current uveitis activity at the time of sampling, elevated DN and CD11c$^-$ DN B cells were observed regardless of whether patients had active or inactive eye inflammation. In line with previously published research, other factors influencing the levels of CD11c$^-$DN1-like B

cells included methotrexate treatment, 'polygo' JIA subtype, active joint count and ethnicity. The association between DN B cells and ethnicity has also been previously noted, with studies in SLE showing a DN2 B cell expansion in African American individuals[41]. Although ANA status was not a significant factor in this model, previous reports have associated positive ANA status with an expansion of DN B cells in JIA[22]. However, this study did not report uveitis status. Thirdly, both CD11c$^-$ DN B cells and memory B cells were among the five variables an unsupervised penalised LASSO regression model determined to be important in identifying JIA patients with uveitis, supporting the notion that these B cell subtypes are potentially key to JIA-uveitis

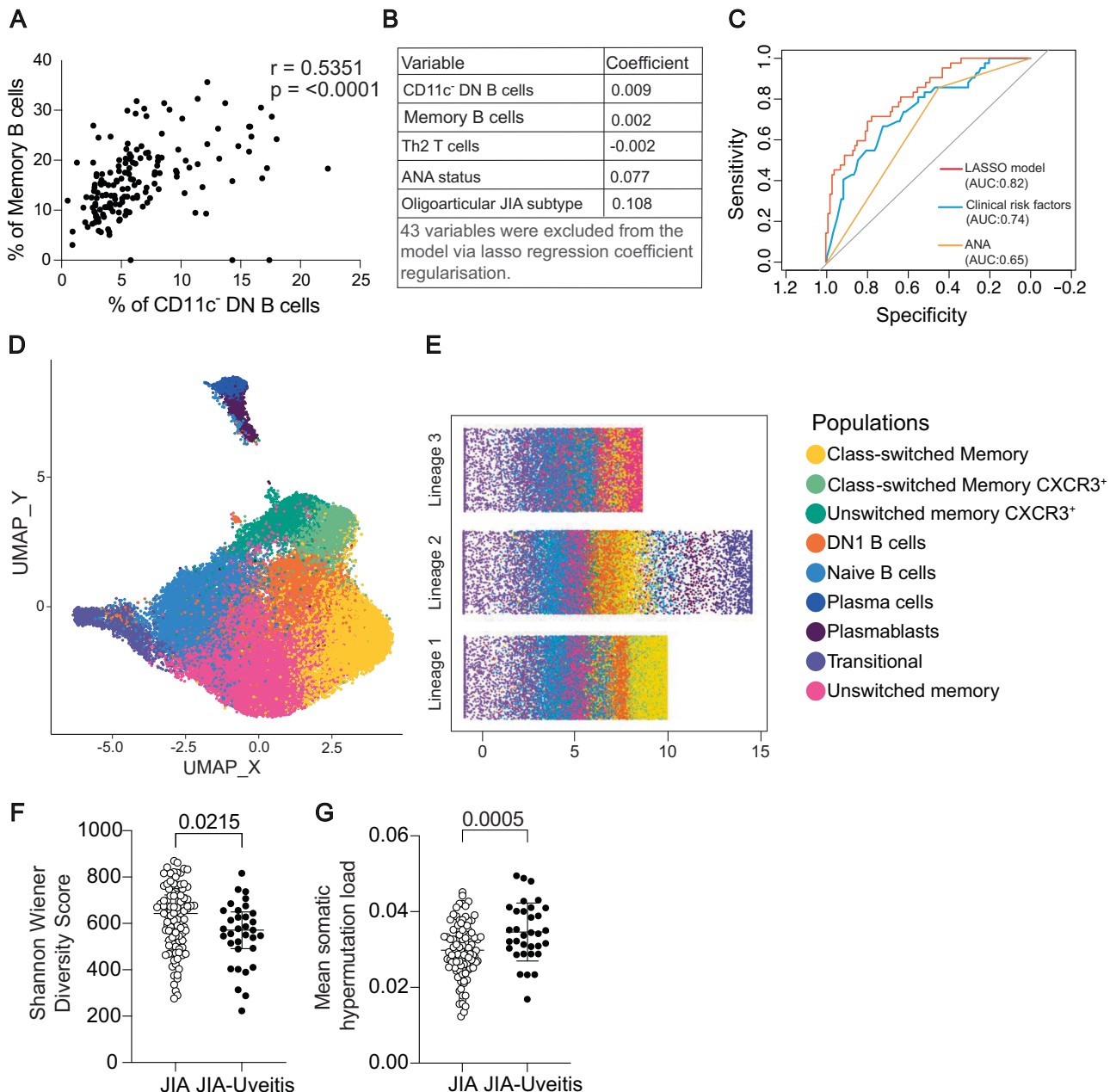

**Fig. 3 | The expansion of DN1 B cells in JIA-uveitis is linked to a concomitant expansion of memory B cells and a more clonal BCR repertoire.** Data from (**A**–**E**) were generated from PBMC from JIA patients without uveitis ($n = 116$) and with uveitis (JIA-Uveitis, $n = 44$). **A** Correlation between Memory B cells and CD11c⁻ DN B cells (two-tailed Spearman correlation, $r = 0.54$, $p < 0.0001$). **B** LASSO regression identifying factors associated with uveitis: CD11c- DN B cells, Memory B cells, Th2 T cells, ANA status and Oligoarticular JIA subtype. 43 variables excluded by the model. **C** ROC curves: LASSO regression (AUC:0.82), multiple logistic regression (AUC:0.74), ANA only (AUC:0.65). AUC values and 95% confidence intervals were calculated using the pROC package in R. **D** U-MAP of CD19 + CD11c- cells from JIA-Uveitis ($n = 9$) vs JIA ($n = 10$). **E** Pseudotime analysis showing 3 B cell trajectories from transitional origin: Lineage 1 (transitional→naïve→CXCR3+ unswitched memory→DN1→class-switched memory CXCR3⁺→class-switched memory), Lineage 2 (transitional→naïve→CXCR3⁺ unswitched memory→DN1→plasmablasts→plasma cells), Lineage 3 (transitional→naïve→CXCR3⁺ unswitched memory→unswitched memory). **F** Shannon Wiener diversity scores of BCR repertoire from bulk B cell RNA-seq of CD19⁺ cells. Each data point represents one patient (biological replicate); JIA $n = 100$, JIA-uveitis $n = 33$. **G** Mean somatic hypermutation load of BCR repertoire from bulk B cells RNA-seq of CD19⁺. Each data point represents one patient (biological replicate); JIA $n = 98$, JIA-uveitis $n = 33$. The effect of uveitis status on Shannon wiener diversity (**F**) mean somatic hypermutation load (**G**) (per sample) was tested using a multiple linear regression model (ordinary least squares), controlling for age and sex. For (**F** and **G**), $p$-values were derived from two-sided $t$ tests of the regression coefficients. Error bars representing median ± IQR.

pathology. This new model was able to segregate JIA-uveitis from JIA patients with an area under the curve (AUC) of 0.82, which is higher than a model using only known clinical risk factors (AUC: 0.74)[29,31]. These findings could be clinically important as an expansion of CD11c⁻ DN and memory B cells in the blood of JIA patients could be used to identify when patients are undergoing an active flare of uveitis during routine rheumatology appointments, prompting an urgent

ophthalmology referral. This is particularly important as even at their highest frequency, uveitis screening only occurs once every 2 months[42,43]. In asymptomatic or younger children who cannot voice any changes to vision or pain, 2 months of uncontrolled eye inflammation may be enough to cause significant damage and permanent vision loss[6]. It is important to note that risk factors for uveitis development in JIA, such as ILAR subtype and ANA positivity, are in no way

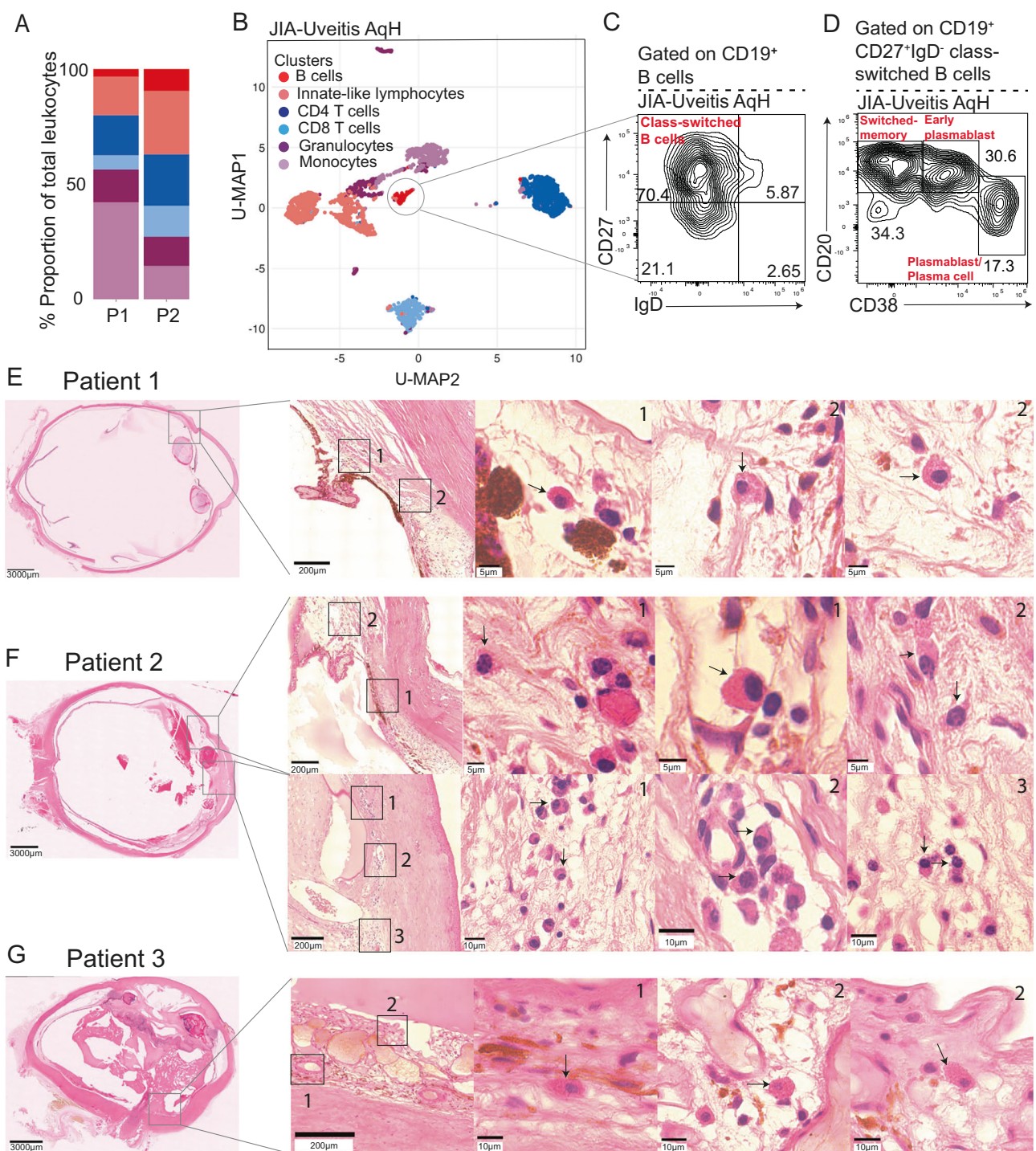

**Fig. 4 | B cells and plasma cells infiltrate the ocular compartment in JIA-uveitis patients. A** Bar chart showing the proportion of each U-MAP cluster as a proportion of total leucocytes infiltrating the AqH in n = 2 JIA-uveitis patients. **B** U-MAP showing the AqH infiltrating leucocytes in JIA-Uveitis patients (n = 2), a grey circle identifies the B cell cluster. Representative flow cytometry plots showing the frequency of CD19⁺ B cells expressing (**C**) CD27 and IgD, and (**D**) CD38 and CD20 within the AqH of n = 1 JIA-uveitis patient. **E**−**G**, H & E staining of historical whole enucleated JIA-uveitis eyes (n = 3, patients 1, 2 & 3). Light grey boxes indicate regions of interest, with magnified views shown in numbered images 1–3. Scale bars are displayed on each image. **E** Patient 1 showed plasma cell infiltration in the iris. **F** Patient 2 showed plasma cell infiltration in the iris and cornea. **G** Patient 3 showed plasma cell infiltration in the choroid. For each patient row, one whole eye section is shown, followed by an image showing numbered areas of plasma cell infiltration and example plasma cell images corresponding to highlighted areas. Black arrows indicate plasma cells identified based on morphology and confirmed by an expert histopathologist.

definitive and new tools are needed to monitor patients across the spectrum of JIA. This approach is supported by our study showing that the DN1 B cell signature associated with uveitis development seemed to span multiple JIA subtypes and recent studies of synovial tissue biopsies in JIA showing that the main drivers of heterogeneity within

inflamed tissues are B cell/plasma cells and myeloid gene signatures, irrespective of ILAR subtype[37].

Despite the described significant phenotypic changes observed in the peripheral B cell compartment of JIA-uveitis and JIA patients, RNAseq transcriptomic analysis found no significant differences in

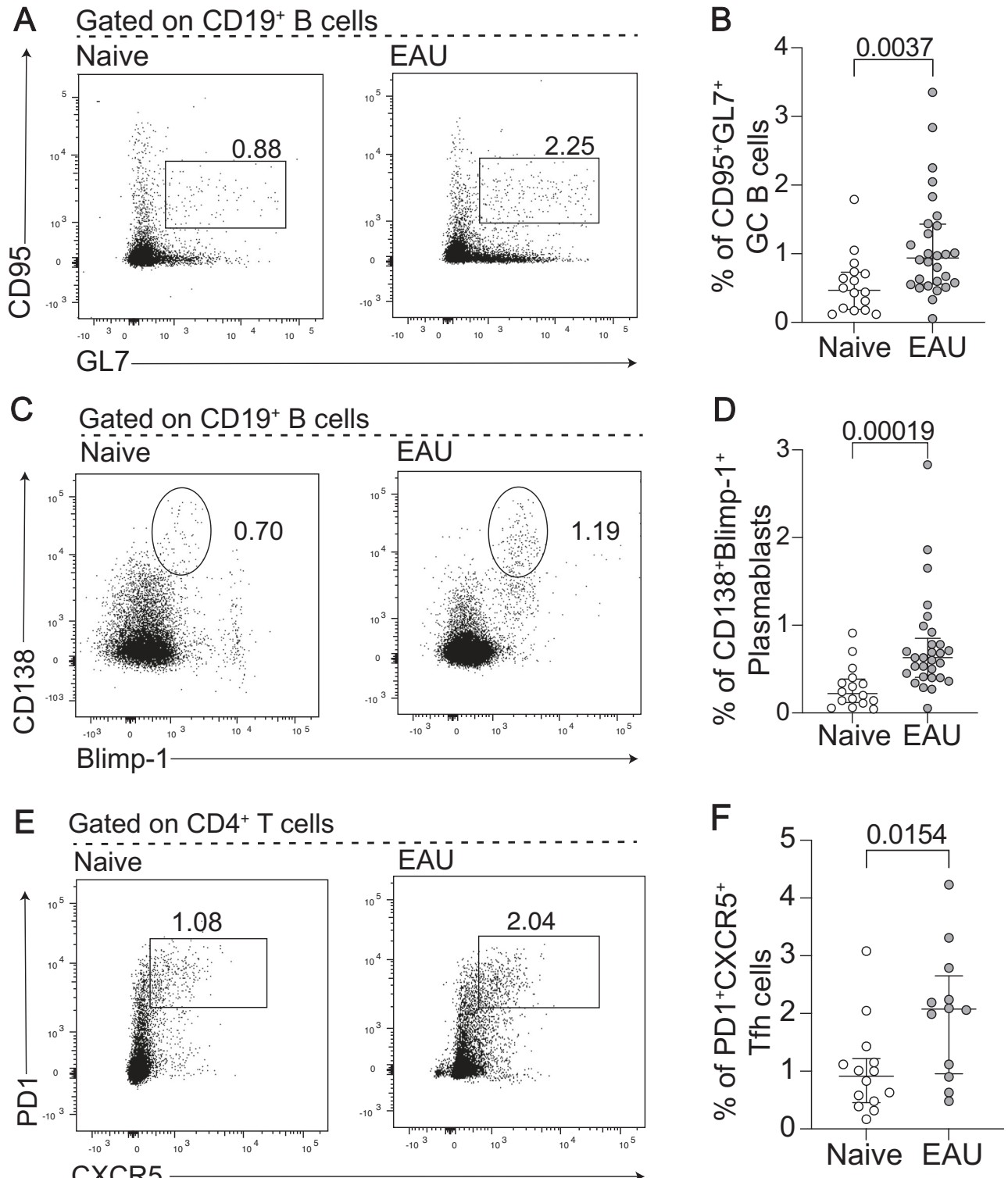

**Fig. 5 | EAU mice show features of peripheral antigen activation and heightened germinal centre reactions in the B cell compartment.** Data for (**A**–**D**) is generated from the spleen of naive mice ($n = 16$) and mice at D21 ($n = 29$) post EAU initiation. Representative flow cytometry plots (**A**) and dot plot (**B**) showing the frequency of CD95+GL7+ Germinal centre (GC) B cells within CD19+ live singlets. Representative flow cytometry plots (**C**) and dot plot (**D**) of CD138+Blimp-1+ plasmablasts within CD19+ live singlets. Representative flow cytometry plots (**E**) and (**F**) dot plot (**F**) showing the frequency of PD1+CXCR5+ T follicular helper (Tfh) T cells within CD4+ live singlets. The significance of the difference between all groups was determined using the two-tailed Mann-Whitney test. *P*-values below or equal to 0.05 are shown on graphs, and the significance threshold was set at ≤ 0.05. Error bars represent median ± IQR.

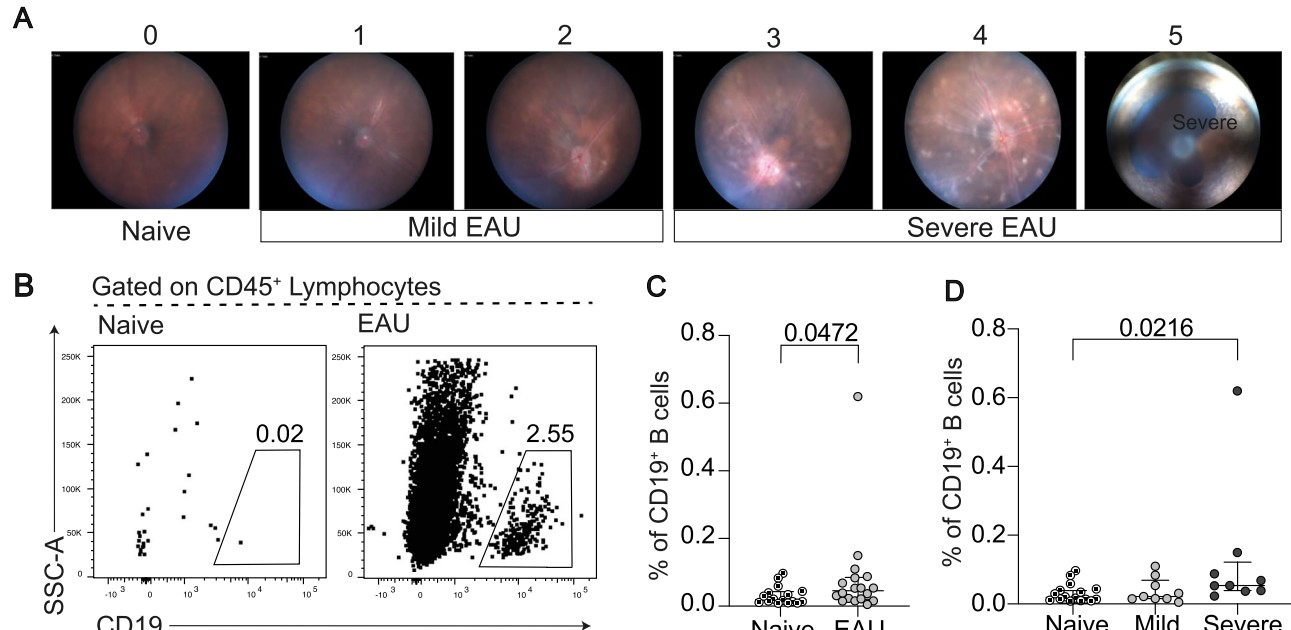

**Fig. 6 | B cells infiltrate the retinas of mice, specifically in those with severe uveitis.** Data were generated from the retina of naive mice ($n = 16$) and mice at D21 ($n = 18$) post EAU initiation. **A** Example fundus images of the adapted scoring system to create naïve (score 0), mild (score 1-2) and severe (score 3-5) disease phenotypes. Representative flow cytometry plots (**B**) and dot plot (**C**) of CD19⁺ B cells within CD45⁺ live singlets within the retina. (**D**) Dot plot showing the frequency of

gene expression of CD19⁺ B cells between these two groups, even when adjusting for sex and age. A study by Wennink et al[44] also found that CD19⁺ B cells in JIA and JIA-uveitis patients are transcriptionally homogeneous, with further deconvolution analysis revealing heterogeneity among memory B cell genes between patients with active JIA-uveitis and those with arthritis alone, similarly to the flow cytometry data phenotyping data described in our study. Wennink et al also showed that the peripheral B cell compartment is dominated by a high proportion of naive B cells, which are known to be relatively transcriptionally quiescent, suggesting that this high proportion of naive B cells may obscure signals from rarer B cell subsets, such as DN B cells[44]. This may provide an explanation as to why limited transcriptional differences were observed in our study. In future studies, to overcome potential 'noise' from more abundant B cell subsets, specific DN1 B cell subset bulk RNAsequencing or single-cell RNAsequencing could be utilised. Despite limited transcriptional differences, in line with the heightened B cell activation indicated by the expansion of DN1 B cells, an assessment of the BCR repertoire using bulk sequencing data on these CD19⁺ B cells showed reduced BCR repertoire diversity in JIA-uveitis patients compared to JIA patients, even when controlling for sex and age. When assessing the mutational load within the BCR repertoire, a readout of somatic hypermutation, we also found that JIA-uveitis patients had significantly higher mutational loads than JIA patients. Since somatic hypermutation occurs primarily within the GC[45], this finding further supports a potential hypothesis where JIA-uveitis patients could exhibit dysregulated GC responses (Fig. 9). However, furthermore, in-depth studies are needed to fine-tune our understanding of the provenance and differentiation trajectory of DN1 and, more globally, all DN B cell subsets. Studies specifically deleting GC B cells, such as in CD23^cre^BCL6^fl/fl mice and investigating the impact on EAU pathology, would also be informative[46].

DN B cell subsets have been previously associated with autoimmunity, with DN2 B cells being the most frequently studied and expanded in conditions including systemic lupus erythematosus, rheumatoid arthritis and juvenile idiopathic arthritis[17,22]. In contrast,

CD19⁺ B cells within CD45⁺ live singlets within the retina of naïve mice versus mice with mild and severe EAU. Significance of difference between groups in (**C**) was determined using the Mann-Whitney test. For (**D**), the significance of the difference between groups was determined using the Kruskal-Wallis test with Dunn's post-hoc test for pairwise comparisons. *P*-values below or equal to 0.05 are shown on graphs, and the significance threshold was set at ≤ 0.05. Error bars represent median ± IQR.

DN1 B cells have been less frequently implicated in autoimmune diseases, with reports limited to their expansion in IgA nephropathy[19] and a trend for an increased ratio of DN1/DN2 B cells primary antiphospholipid syndrome (APS)[20] both of which are hypothetically linked to GC dysfunction. Our study is the first to link DN1 expansion to a childhood-onset autoimmunity. In IgA nephropathy, the increase in DN1 B cells is accompanied by an expansion of switched memory B cells and pathogenic plasmablast populations, suggesting a similar B cell differentiation trajectory to our results in JIA-uveitis. In APS, there is also reduced diversity of the BCR repertoire, which is suggested to be indicative of GC dysfunction. However, in this study, due to the combination of single-cell RNA sequencing (scRNAseq) with BCR repertoire alongside a known auto-antigen (anti-phospholipid), autoreactive B cell clones can also be tracked from the naïve natural repertoire into the switched memory population. Although JIA-uveitis lacks a known autoantigen, and the specific targets of ANA in JIA remain unknown[47], future studies employing similar techniques may help to resolve if similar altered B cell activation pathways are shared amongst these conditions.

Despite being one of the minority subsets, B cells were present within the AqH of JIA-uveitis patients undergoing cataract surgery and were mainly of a class-switched memory and plasmablast/plasma cell phenotype. This was complemented by analyses demonstrating that plasma cells can be found within different inflamed ocular tissues in historically biobanked and enucleated eyes from JIA-uveitis patients. In previous studies utilising iridectomy tissue from childhood uveitis patients, who have uveitis-associated glaucoma, plasma cells can be found within the iris tissue of ANA⁺ patients, which includes patients both with and without a JIA diagnosis[9,10]. The presence of B cells in the AqH of adult uveitis patients has also been shown to vary significantly across individuals, with a recent study showing that B cells were indetectable in some patients but comprised 43% of the total leucocyte population in others[48]. It is important to note that the accessing the ocular compartment remains challenging especially in children which, to date, has prevented the same 'atlas-ing' of the inflammatory

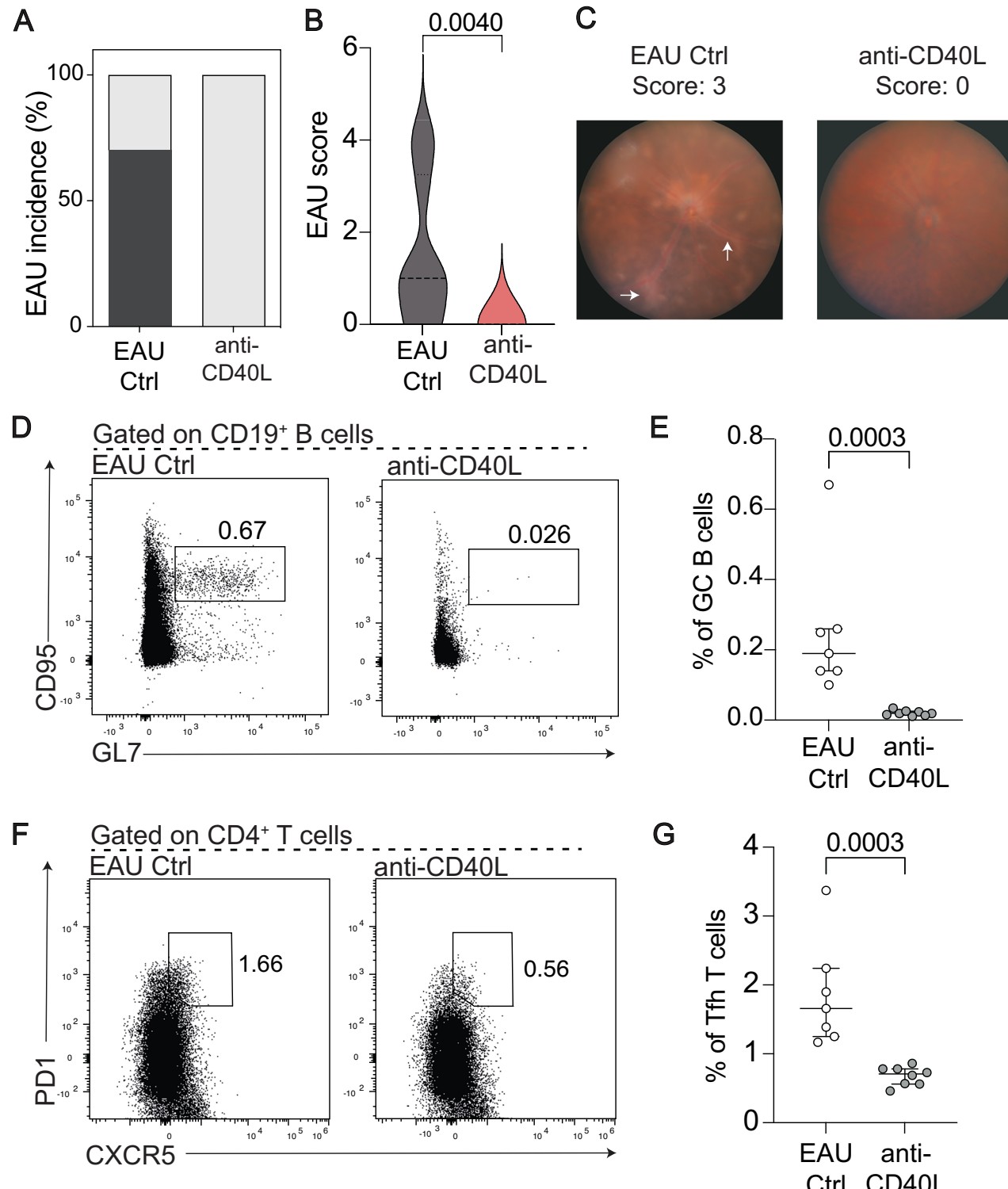

**Fig. 7 | Modulation of B cell:T cell interactions via CD40L antagonism suppresses uveitis severity in vivo.** Data were generated from $n = 7$ EAU Ctrl mice and $n = 8$ EAU mice treated with anti-CD40L every other day from days 4–20 following EAU initiation. **A** Bar plot comparing the incidence of uveitis between standard EAU Ctrl mice (left) and EAU mice treated with anti-CD40L (right). **B** Violin plot showing the retinal EAU disease score at day 21. **C** Example fundoscope images from a EAU Ctrl mouse and an anti-CD40L-treated EAU mouse. **D** Representative flow cytometry plots and (**E**) Dot plot showing the frequency of CD95⁺GL7⁺ germinal centre (GC) B cells within CD19⁺ live singlets. **F** Representative flow cytometry plots and **G.** Dot plot showing the frequency of CXCR5⁺PD1⁺ T follicular helper (Tfh) T cells within CD4⁺ live singlets. Statistical significance between groups was determined using the two-tailed Mann-Whitney test, with $p$-values ≤0.05 shown on graphs. The significance threshold was set at ≤0.05, and error bars represent median ± QR. White arrows point to clinical features of uveitis i.e., swelling of the optic disc and cuffing of vessels.

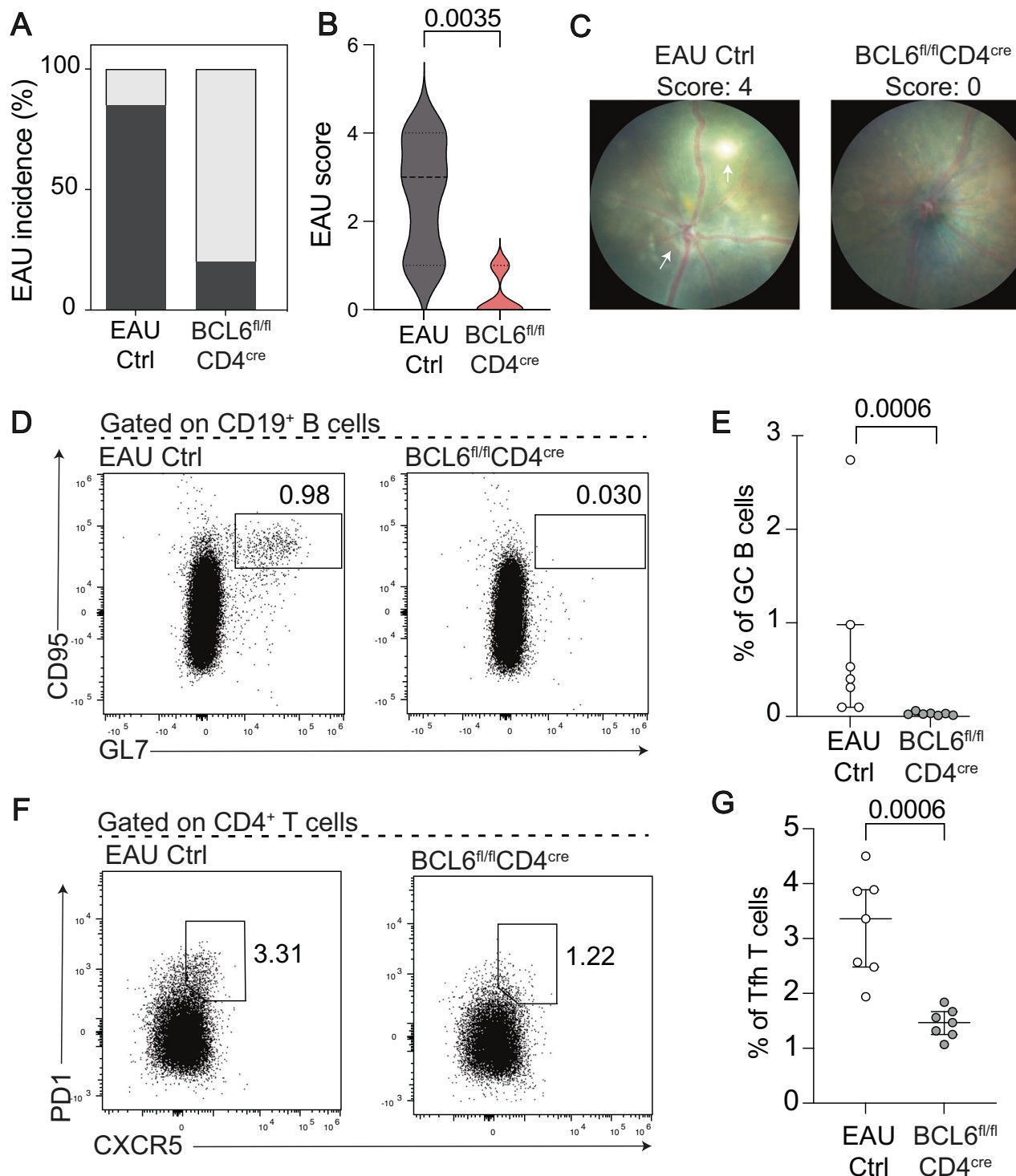

**Fig. 8 | Mice who are deficient in T follicular helper cells have significantly reduced uveitis severity.** Data were generated from two individual experiments with a total of $n = 12$ EAU Ctrl mice and $n = 12$ BCL6$^{fl/fl}$CD4$^{cre}$ mice initiated with EAU. Data shown are representative of one experiment. **A** Bar plot comparing the incidence of uveitis between EAU Ctrl mice (left) and BCL6$^{fl/fl}$CD4$^{cre}$ mice (right). **B** Violin plot showing the retinal EAU disease score at day 21. **C** Example fundoscope images from a EAU Ctrl mouse and a BCL6$^{fl/fl}$CD4$^{cre}$ mouse. **D** Representative flow

cytometry plots and (**E**) Dot plot showing the frequency of CD95$^+$GL7$^+$ germinal centre (GC) B cells within CD19$^+$ live singlets. **F** Representative flow cytometry plots and (**G**). Dot plot showing the frequency of CXCR5$^+$PD1$^+$ Tfh within CD4$^+$ live singlets. Statistical significance between groups was determined using the two-tailed Mann-Whitney test, with $p$-values ≤0.05 shown on graphs. The significance threshold was set at ≤0.05, and error bars represent median ± IQR.

infiltrate of childhood uveitis patients compared to other diseased sites in autoimmune conditions. Future studies performing scRNAseq on AqH cells would be technically challenging but are critical for addressing this gap. Of note, class-switched memory B cells and

plasmablasts/plasma cells have been previously observed in the synovial fluid extracted from inflamed JIA joints[22]. Thus, antigen-activated B cell subsets can therefore be found in both the ocular and synovial compartments of JIA and JIA-uveitis patients. Notably, within

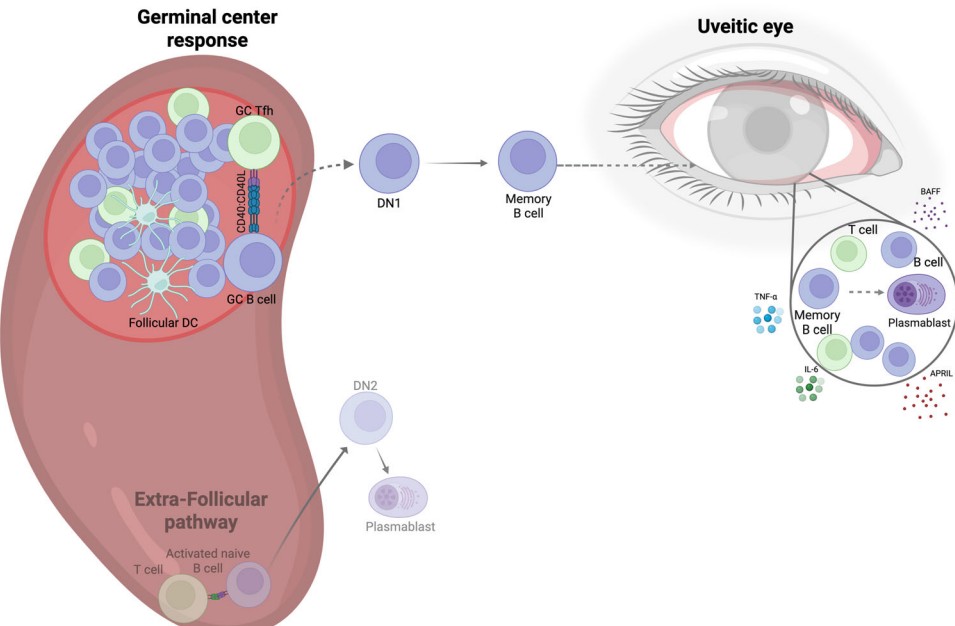

**Fig. 9 | Proposed schematic showing new hypothesis for how dysregulated B and T cell interactions could contribute to pathogenesis of JIA-uveitis.** Altered B-cell and T-cell interactions could lead to increased GC activation in secondary lymphoid organs, which could lead to an expansion of DN1 B cells in the blood of JIA-uveitis patients. These cells then differentiate into memory B cells, which could then migrate to the uveitic eye and cross the blood-retina barrier. Once in the eye, B cells could differentiate into antibody-producing plasma cells contributing to the general inflammatory milieu and drive chronic inflammation. Future experiments are needed to confirm both the provenance of DN1 B cells and the exact role of GC B cells in EAU pathology. Alternative DN B cell subset trajectories are shown greyed out. Created in BioRender. Jebson, B. (2026) https://BioRender.com/s4ga0th.

the synovial tissue of JIA patients, several studies have shown that there is plasma cell infiltration in the tissue and that this infiltration is associated with a worse arthritic trajectory[35,36]. Early identification of JIA-uveitis patients with ocular plasma cell infiltration may offer the ability to intervene with more targeted therapies before the development of severe sight-threatening complications such as cataracts or glaucoma. However, the relationship between autoantibody positivity in arthritis and B-cell involvement in uveitis is likely to be complex, as shown through patients with RF+ polyarticular arthritis who respond well to rituximab treatment[49,50], but appear to be paradoxically protected from uveitis development[51]. This suggests that different B cell subsets or activation states may have distinct roles in the pathogenesis of joint versus ocular inflammation, and that perhaps it is the outcome of the interaction between B cells with other immune cells such as T cells that conditions whether B cells contribute to uveitis pathogenesis. Supporting this concept of coordinated immune cell interaction, a recent large-scale GWAS study from the CLUSTER consortium identified that in JIA-uveitis patients, the top genetic risk factors were located within the HLA region (*HLA-DRB1, HLA, DPB1 and HLA-A*), indicating an important role for T cells and their interactions with antigen-presenting cells such as B cells in disease pathology[52].

There is no animal model which recapitulates all the features of JIA-uveitis, where there is concomitant joint and eye inflammation in juvenile animals, leading to some barriers regarding therapeutic innovation in JIA-uveitis. To address this, we assessed whether we could use the most widely used animal model of uveitis, EAU, to model some features of altered B-cell activation observed in JIA-uveitis. Although this model is predominantly considered to be T cell driven, previous studies have also demonstrated important roles for B cells in EAU[53–55], including that depletion of B cells in EAU post-disease onset suppresses EAU severity, whilst administration of anti-CD20 prior-disease onset has no effect[12]. In our study, EAU induction was associated with increased GC B cells, plasmablasts and Tfh cells and infiltration of B cells into the ocular compartment of mice with severe

disease phenotypes. Our study is not the first to show that in animal models of uveitis, there is also a strong association between disease severity/chronicity and B-cell infiltration into the ocular compartment. In mice with EAU, induced by injection of interphotoreceptor binding protein (IRBP) and eventual loss of tolerance to retinal antigens, there is increased infiltration of B220+ B cells into the retina at days 38-43 compared to days 24-26 post-disease induction[4]. In addition, in *Aire-/-* mice, which develop multi-system autoimmunity including uveitis, there is a direct positive correlation between disease score and infiltration of B cells and plasma cells into the retina, where there is development of tertiary lymphoid structures (TLS)[56]. Retinal TLS also develop in the R161H TCR transgenic model of spontaneous EAU. Although R161H TLS are initially associated with slower loss of visual function, the presence of TLSs with abundant plasma cells is ultimately associated with greater retinal damage[57]. It has been hypothesised that the worse outcomes associated with B-cell infiltration and late-stage tertiary lymphoid structure TLS formation is due to the local production of autoantibodies[57].

Although it has been previously published that anti-CD20 administration in EAU suppresses uveitis severity[12] and that the anti-CD20 B cell depletion therapy rituximab has shown some efficacy in case studies of severe non-responsive JIA-uveitis[58], it is not considered a mainstream treatment. In addition, the use of rituximab in other autoimmune diseases is not without limitations due to its variable tissue penetrance and inability to deplete plasma cells[59,60]. Given that plasma cell depletion with bortezimab significantly reduces EAU severity[12,61], and that CD19-targeted CAR T cells are being explored in treatment-resistant SLE to deplete the entire B cell population, including plasma cells[62], we hypothesised that targeting B:T cell interactions might be a more effective novel therapeutic strategy than a sporadic B cell depletion. Accordingly, our data shows that anti-CD40L antagonism has a significant effect on the severity of EAU. Since CD40L has broad immune effects, including the activation of CD8+ T cells[63], we also employed a more targeted, complementary genetic

approach using BCL6[fl/fl]CD4[cre] mice, which specifically lack BCL6 expression within T cells and therefore cannot develop Tfh cells, a crucial B helper subset[27]. The dramatic reduction in EAU severity we observed in these mice suggests that Tfh cells are critical for disease pathology and that targeted disruption of B:T cell interactions is sufficient to ameliorate uveitis. Importantly, these findings could have therapeutic relevance. After early trials using anti-CD40L monoclonal antibodies were halted due to thrombotic side-effects, second-generation CD40L monoclonal antibodies such as Frexalimab and Dapirolizumab pegol (DZP) have shown disease modifying properties in disorders such as multiple sclerosis (MS)[64] and SLE respectively[28]. Performing similar studies in spontaneous experimental models of uveitis could provide further evidence that anti-CD40L antagonism, such as the strategies currently being employed in MS and SLE, may be efficacious in suppressing ocular inflammation in JIA-uveitis and potentially other forms of child and adult-onset uveitic disease.

Our study is not without its limitations. The cohort of JIA patients included in this study are a mix of all JIA subtypes (other than systemic JIA) at different points in their disease course and currently undertaking or have previously been on various treatment regimens - meaning they are relatively heterogeneous with respect to these factors. Genetic and transcriptional JIA studies show that ILAR category and Methotrexate/anti-TNFα agent treatment can alter the phenotypic and transcriptional profile of immune cells isolated from these patients[65,66]. Another potential confounder of these data is that all the patients included were recruited to cohort studies either at the time of a joint injection, or before beginning a new treatment for their arthritis. As a result, they all have 'active' JIA or joint inflammation at the time of sample collection, and it is unclear whether this overrides any potential differences associated with uveitic disease. In addition, as the included patients were recruited for arthritis studies, there was limited uveitis-specific clinical information available. Key information, such as uveitis severity and response to treatment, would be beneficial for correlating with identified B cell activation markers, further strengthening the clinical utility of these findings. Missing clinical data was also an issue with the historical archival enucleated eye tissues. For future JIA-uveitis studies, we aim to recruit uveitis patients at diagnosis and follow them longitudinally with comprehensive clinical information to determine whether this altered B cell activation profile represents a feature of active disease or a risk marker for uveitis development. As noted, due to the challenging nature of accessing the ocular site, we were only able to assess B-cell infiltration in JIA-uveitis with severe disease phenotypes and unable to assess whether B cells were present in new-onset cases. Finally, the most common uveitis in JIA-uveitis patients is chronic anterior uveitis, whilst our experimental model of choice, EAU, mostly results in pan-uveitis. In this model, eye inflammation is also not accompanied by any joint inflammation. Although there are mouse models of uveitis which are comorbid with arthritis, these models focus on the anterior forms of uveitis associated with HLA-B27 positivity and spondyloarthropathy[67]. Regarding B cell involvement, the choroidal layer, which has been shown to house ectopic lymphoid structures in human uveitis[68], is much thinner in mice and the immunoglobulins produced by mice are significantly different to the human counterparts. Despite its limitations, EAU has informed various therapeutic advances for uveitis, including the use of anti-TNFα and anti-IFNα therapies for seronegative spondyloarthropathies and Behcet's disease-associated uveitis[69–71]. Moving away from in vivo studies, uveal organoids may offer a more physiologically relevant platform to study uveitis in the future. However, current ocular organoid systems primarily model retinal structures rather than the uvea (iris, ciliary body and choroid), which is the part of the eye most affected in JIA-uveitis. Any future uveal organoid would need to incorporate immune cell infiltration to fully model this inflammatory condition.

Despite decades long knowledge that the presence of anti-nucleic antibodies is a key risk factor for uveitis development in JIA patients, there remains an underappreciation of the role that antibody-producing cells and B cells play in JIA-uveitis pathogenesis. In this study, we provide evidence that JIA-uveitis patients can be distinguished from JIA patients by features of enhanced B cell activation and potentially altered B:T cell interactions, which include an expansion of DN1 B cells, developmentally linked memory B cells and a clonally expanded BCR repertoire with enhanced levels of somatic hypermutation. This is accompanied by ocular-infiltration of B cells and particularly plasma cells in JIA-uveitis patients with structural synechiae driven by severe ocular inflammation. Demonstrating the translational relevance of these findings, we found that anti-CD40L antagonism or genetic deficiency of Tfh helper cells significantly suppressed experimental uveitis severity. These data suggest that dysregulated interactions between T cells (particularly Tfh helper cells) and B cells may be a critical pathogenic mechanism driving ocular inflammation in JIA-uveitis patients. These data offer a conceptual shift that T cells as the sole drivers of pathology in JIA-uveitis and show the importance of coordinated B cell:T cell responses, and potentially those that occur within GC. Considering that only one NICE-approved biologic therapy currently exists for JIA-uveitis patients (Adalimumab)[3,42], any scientific evidence supporting targeted disruption of pathogenic B cell:T cell interactions could guide the repurposing of existing therapeutics or initiation of new clinical trials to improve patient outcomes.

## Methods

### Human participants
JIA and JIA-uveitis patients were recruited through three main pathways, through the CLUSTER consortium (https://www.clusterconsortium.org.uk/), Childhood Ocular Inflammatory Disease (**CHOIR**) Research Tissue biobank (https://www.hra.nhs.uk/planning-and-improving-research/application-summaries/research-summaries/choir-biobank/), and via the Moorfield Pathology Diagnosis Archive. All research involving human participants was conducted in accordance with the declaration of Helsinki and was approved by the relevant institutional and national research committees. Written informed consent was obtained for all samples used in this study.

### CLUSTER consortium cohort
Patients within the CLUSTER consortium were recruited at Great between 1999 and 2019 from Great Ormond Street Hospital (GOSH) London, UK with parent or legal guardian consent and age appropriate assent for paediatric participants. Recruitment, PBMC and SFMC sampling was approved by the London-Bloomsbury Research Ethics (reference: 95RU04 & 04RU07). Patient demographics can be found in Table 1. Please note, for patient ancestry data the original terminology as recorded in source studies has been reported to ensure data integrity and avoid any misclassification. Where possible, all demographic data has been reported, and the proportion of missing data (such as HLA-B27 status) is declared in the demographics tables.

### CHOIR biobank
All patients within the CHOIR Biobank were recruited between 2023 and 2024 to GOSH, London, UK with parent or legal guardian consent and age-appropriate assent for paediatric participants. Study recruitment was approved by the London-Bloomsbury Research Ethics (reference22:/LO/0575). Patient demographics can be found in Supplementary Table 5.

### Moorfields pathology diagnostic archive
Enucleated eyes obtained as part of standard clinical care from JIA-associated uveitis patients were provided via Moorfields Eye Hospital Biobank under the approval of South-West Central Bristol Research Ethics Committee (Ethics reference20:/SW/0031-2022ETR84). All donors provided written informed consent for biobanking and

 

approved research use. Use of these samples in the present study is covered by the ethical approvals listed above. Patient demographics can be found in Supplementary Table 6.

## Human sample collection (blood, synovial fluid and aqueous humour)

Venous blood for PBMCs was collected in PFH (preservative-free heparin) coated monovettes (Sarstedt, cat no: 03.1628.100). Synovial fluid was collected in 20 ml falcons containing PFH following arthrocentesis. Aqueous humour (AqH) was collected in sterile 1.5 ml Eppendorf's containing 200ul PBS (Sigma-Aldrich, cat no: P4474) with EDTA (2 mM, Sigma Aldrich, cat no: 574795). Peripheral blood mononuclear cells (PBMCs) were isolated via density centrifugation as previously described[72]. Synovial fluid mononuclear cells (SFMCs) were processed identically as PBMC except they were mixed 1:1 with complete media (RPMI-1640 supplemented with glutamine, penicillin and streptomycin (Sigma Aldrich, cat no: 12352207) and 10% Foetal Calf Serum (Gibco, cat no: A5256701) and treated with Hyaluronidase (10 µ/ml, Sigma-Aldrich, cat no: HX0514) for 30 min at 37 °C prior to density centrifugation. Following density centrifugation, PBMC and SFMC were cryopreserved at −196 °C in freezing media (10% Dimethyl sulfoxide (DMSO, Sigma-Aldrich, cat no:D2653), 90% Foetal Calf Serum (FCS, Gibco) and stored until use. AqH samples were collected in sterile tubes and arrived on ice. Immediately after receival, AqH samples were centrifuged at 300xg for 10 min and processed for analysis.

## H&E histological analysis of enucleated eye tissue

JIA-uveitis eyes were sourced from formalin-fixed, paraffin-embedded whole enucleated eyes in blocks created between 1998 and 2000 and obtained from Moorfields Biobank (Ethics reference 20/SW/0031). FFPE sections were cut at 4 µm thick and stained with haematoxylin and eosin. Sections were imaged using a Thunder Imager Live Cell microscope (Leica Microsystems) with a 5 x objective for the whole eye overview and 63 x objective in areas of interest. Image processing was carried out using Fiji ImageJ[73], with plasma cells identified morphologically and confirmed by an expert clinical histopathologist.

## Sex as a biological variable

Human studies included both male and female participants. The influence of sex on immunophenotyping data was explored using linear and LASSO regression models. For all transcriptomic analyses, sex and age were treated as biological variables, and data were adjusted accordingly. In mouse studies, female mice were used for all experiments according to standard protocols for the EAU model (66). Females are preferentially used in EAU experiments as uveitis, including JIA-uveitis, is more prevalent in the females versus male human population.

## Animal strains and husbandry

Female C57BL/6 mice were purchased from Envigo at 6 weeks old (stock no: 000664). Experiments were initiated when mice were between 6 and 8 weeks of age, unless otherwise stated and housed in UCL biological service units. Mice were kept in individually ventilated specific pathogen-free cages in a controlled environment with a 12 h light/dark cycle and a temperature of 22 +/− 2 °C. All mice were fed a standard diet and had access to water ad libitum. Control mice were housed in separate cages to prevent inadvertent exposure to IRBP and CFA used for EAU induction. For all animal experiments, our previous studies have demonstrated that a minimum group size of 8 is needed for a significance level of 0.05 (5%), power of 80%, and an estimated effect size of approximately 10% to identify differences between groups. For all experiments where mice were given treatments, mice were randomised, and the handler was blinded. Mice were euthanised using Schedule 1 methods. All experiments were approved by the Animal Welfare and Ethical Review Body of University College London and authorised by the United Kingdom Home Office. All animal experiments were performed in accordance with the ARVO statement of 'The Use of Animals in Ophthalmic and Vision Research'. BCL6$^{fl/fl}$CD4$^{cre}$ mice were kindly donated by Professor Richard Jenner, UCL. Briefly, parental strains (BCL6$^{fl/fl}$ mice, Jackson strain no: 023727) and (CD4$^{cre}$ mice, Jackson strain no: 022071) were bred to give BCL6$^{fl/fl}$CD4$^{cre}$ mice as previously described[27]. EAU was initiated in female BCL6$^{fl/fl}$CD4$^{cre}$ mice which were between 6-8 weeks old. For EAU controls' C57BL/6 or BCL6$^{fl/fl}$ mice were used as appropriate.

## Induction and clinical evaluation of experimental autoimmune uveitis (EAU)

IRBP 1-20 (peptide sequence: GPTHLFQPSLVLDMAKVLLD) was purchased and synthesised by Cambridge peptides, UK (cat no: HY-P1861A-10mg). IRBP 1–20 was reconstituted in 10% DMSO in PBS (Sigma Aldrich can no: P4474 and D2653) to achieve a final concentration of 20 mg/ml and stored at −80 °C prior to use. On the day of EAU initiation, a stock solution containing of IRBP antigen was prepared by diluting with PBS to give a concentration of 10 mg/ml and mixed 1:1 with Complete Freund's Adjuvant (Sigma Aldrich, cat no: F5881) supplemented with 1.5 mg/ml Mycobacterium tuberculosis (Difco, cat no: 231141). 50 µl of IRBP CFA mix was injected subcutaneously (SC) into the right and left flanks of the mouse – resulting in a total dose of 100 µl at 500 mg per mouse. Mice also received an intraperitoneal injection of 1.5 mg of Bordetella Pertussis Toxin (Tocris, cat no: 3097), which was diluted in PBS to make a 0.1 mg/ml stock. Humane endpoints were predefined according to the institutional ethical approval. Specifically, any mouse that lost more than 15% of its pre-procedure body weight, showed dyspnoea, ruffled fur, weakness, dehydration, persistent hunching, or exhibited ulcers >3 mm at the injection site or infection not resolving within 48 h was immediately euthanised by a Schedule 1 method. Mice were scored based on fundus images obtained using a Micron III or Micron V imaging system (Pheonix Research Labs). The scoring system was adapted based on the previously published method[74]. Briefly, images were scored by evaluating the ocular inflammation based on the following metrics: 1, Optic disc swelling (margin of the optic disc becomes blurred as inflammation occurs); 2, Retinal vasculitis (engorged vessels with cuffing of white infiltrates around edges); 3, Retinal tissue infiltrates (white lesions that occur separately from vessels); 4, Structural damage (retinal atrophy with scarring); 5, Posterior Synechiae. All experiments were terminated on day 21 post-disease induction.

## Disruption of B:T cell interactions using CD40L antibody

Anti-CD40L (clone:MR-1; IgG) and IgG isotype control antibodies were purchased from BioXcell (cat no: BE0017-1 and BE0091). Four days after the initiation of EAU, C57BL/6 mice were injected intraperitoneally (I.P) three times per week (Monday – Wednesday – Friday) with 500 µg of anti-CD40L or isotype as a control. Mice were imaged and euthanised on day-21 post induction.

## Dissection of mouse tissues and sample preparation

The retina and spleen were taken from sacrificed mice for further analysis. Briefly, mice were enucleated, and the retina was removed using a dissecting microscope. Retinal samples were manually dissociated. Spleens were removed and placed into complete media. Splenocytes were disaggregated by pushing through a 70µm cell strainer (Sigma Aldrich, cat no: CLS352340). For spleen samples, red blood cells were lysed using lysis buffer (Sigma Aldrich, cat no: R7757), and then lysis was stopped by washing in RPMI 1640 media supplemented with 10% FCS (Sigma Aldrich, RPMI cat no 12352207 and Gibco FCS cat no: A5256701). Both tissues were then processed for downstream flow cytometry analysis.

## Flow cytometry

Flow cytometry protocols were performed identically for both human and mouse samples and for both conventional and spectral flow cytometry unless otherwise stated. Isolated cells were stained for multi-colour flow cytometry with fluorochrome-conjugated antibodies as previously described[75]. Briefly, cells were plated in a 96-well round bottom plate and stained with 50 ml viability dye Live/Dead fixable blue stain or Zombie NIR viability dye (Biolegend, cat no: 423105). Cells were then washed and stained for surface markers with fluorochrome-conjugated antibodies and fixed by incubation in 2% paraformaldehyde (PFA, Sigma Aldrich, cat no: 394513). For intracellular staining, instead of PFA, cells were incubated with Transcription Factor Staining Buffer (eBiosciences, cat no: 00-5523-00). Cells were then washed and incubated with fluorochrome-conjugated antibodies in perm buffer (eBiosciences, cat no: 00-5523-00). Following staining, all samples were resuspended in 200 μl FACS buffer for analysis on an LSR II cell analyser (BD Biosciences) or Sony spectral cell analyser ID7000 (Sony Biotechnology). For the details of antibodies used for human standard panels, see Supplementary Table 7; for mouse standard panels see Supplementary Table 8; for human spectral panels see Supplementary Table 9. Flow cytometric data were analysed using FlowJo version 10 and R studio version 4.4.1. Gating strategies can be found in Supplementary Figs. 8–11.

## Cell sorting, RNA isolation and library preparation

Cells were stained with fluorochrome-conjugated antibodies (Supplementary Table 10) in MACs buffer (PBS, 0.5% FCS, 0.5 mM EDTA) and filtered through polypropylene tubes with 70 m filter caps (Falcon, cat no: 352235) and DAPI, (Thermo Fischer Scientific, cat no: D1306) was added to each sample. CD19[+] cells were sorted using a FACS Aria (BD Biosciences) into 1.5 ml RNA- free Eppendorf's containing 100 ml RNAse free PBS. Purity cheques were performed on sorted populations and reached 98% purity. Sorted cells were kept on ice and immediately processed for RNA isolation to prevent RNA degradation. On average 330,976 B cells were sorted per sample, though this varied considerably (range: 42,125 – 2,592,752). RNA was isolated from sorted cell populations using the PicoPure kit (Thermo Fisher, cat no: KIT0204) according to the manufacturer's instructions. The quality of RNA was then assessed using a NanoDrop spectrophotometer, and samples with a Nanodrop scored were submitted to UCL Genomics Facility. Total RNA quantification and integrity was confirmed using Agilent's 4200 Tapestation (Standard Total RNA assay). RINs values were confirmed to all be >7.0, indicating good to high integrity RNA suitable for library preparation. For each sample, 50 ng of total RNA were processed using a commercial mRNA library preparation kit according to the manufacturer's instructions. Two different kits were used: the KAPA mRNA HyperPrep Kit (Roche, cat no: KK8580) and the TruSeq Stranded mRNA Library prep kit (Illumina, cat no: 20020492).

## RNA sequencing and data analysis

High yield, adaptor-dimer free libraries were confirmed on the Agilent TapeStation 4200 (High Sensitivity Agilent DNA 1000 assay). Samples were quantified using the Qubit High Sensitivity DNA assay and normalised to 10 nM. An equal volume of each library was pooled together and re-quantified by Qubit. Samples were sequenced on the NovaSeq instrument (Illumina, San Diego, US) at 300pM, using a 100 bp paired read run with corresponding 8 bp dual sample index and 8 bp unique molecular index reads. Run data were demultiplexed and converted to FASTQ files using Illumina's BCL Convert Software v3.75. At the same time, the unique molecular index was moved to the read header for downstream analysis. FASTQ files were then aligned to the human genome UCSC hg38 using STAR software[76]. Aligned reads were then UMI deduplicated using Je-suite (2.0.2.RC)[77] and reads per transcript were counted by FeatureCounts[78] in order to produce a digital output of gene expression. Differential expression analysis was performed using the R package DESeq2[79]. All annotation and sequences were obtained from Illumina iGenomes (http://emea.support.illumina.com/sequencing/sequencing_software/igenome.html). Genes were considered differentially expressed if they had an adjusted p-value of less than 0.05 and a log2fold change greater than 1.

## B cell receptor repertoire analysis

B cell receptor (BCR) repertoire analysis was performed on bulk CD19[+] RNA sequencing data using the MiXCR software (v4.5.0)[34,80]. The MiXCR analyse command was used to execute its optimised pipeline for the analysis of bulk RNA sequencing data. This pipeline involves: 1. The alignment of sequencing reads from transcripts to immunoglobulin light and heavy genes. 2. The assembly of reads, containing only fragments of CDR3, into longer contigs covering the full or nearly full CDR3 region. 3. Clustering assembled sequences to the clonotypes based on their CDR3 region. 4. The export of clonotypes and corresponding abundance information. The MiXCR individual post-analysis module was then used for downsampling, normalisation and calculation of the CDR3 diversity and gene segment data for each sample. BCR clonotype data was tested for differences between samples from JIA and JIA-uveitis patients. The effects of uveitis status on BCR clonality (Shannon-weiner diversity), CDR3 amino acid length, individual isotype usages, and individual IGHV gene usages were tested separately using multiple linear regression models (ordinary least squares), controlling for age and sex. P-values for each were derived from two-sided $t$ tests of the regression coefficients. For the sake of displaying differences in IGHV gene usage, where some genes are not found in some samples, a pseudocount equal to the minimum proportion seen in the table was added, then log transformed the data. Log transformation means that the coefficients estimated by the linear model can be interpreted as estimations of log fold change.

## B cell receptor somatic hypermutation analysis

Assembled contigs containing the full CDR3 region, which are output by the MiXCR pipeline, were converted into.fasta format, which were used as input for the standard Immcantation change-O pathway: assigngenes.py, then MakeDB.py, then CreateGermlines.py. We have included a Bash script with details on this linking between the MiXCR and Immcantation pipelines (mixcr_to_immcantation_script.sh) within the supplementary data file 1. Following this, the observedMutations function from the Immcantation shazam package was used to calculate mutational frequencies in the V segment for each contig in RStudioTM. For each sample, the mean of mutational frequences across all heavy chain contigs was calculated. Only contigs assembled from more than one read were included in the calculation of the mean mutational frequency. The effect of uveitis status on mean mutational load (per sample) was tested using a multiple linear regression model (ordinary least squares), controlling for age and sex. A $p$-value was derived from a two-sided $t$ test of the regression coefficient.

## Unsupervised clustering

Unsupervised clustering analysis was performed using two different R packages. For human aqueous humour analysis, the CATALYST package was used to identify 6 different clusters using FlowSOM (6037 cells total) from $n = 2$ patients[81]. For JIA and JIA-Uveitis peripheral blood mononuclear cells, the Spectre R package was used[82]. Briefly, 19 fcs files from CD11c[-]CD19[+] B cells were clustered using Spectre and FlowSOM to identify 15 different B cell clusters. Phenotypically similar clusters were manually merged to avoid over-clustering, resulting in 9 individual populations (38976 cells total).

## Pseudotime analysis

The 9 populations of CD11c[-]CD19[+] B cell clusters were analysed using the slingshot package[83]. Transitional B cells were specified as the initial cluster, but no endpoint was provided. Visualisation was achieved by

plotting pseudotime values across all cell subsets within the 3 inferred lineages using ggplot.

## Data visualisation

Plots were generated using GraphPad Prism (version 10.2.1) and R. In R, visualisations were created with the ggplot2 and clusterProfiler packages. Schematic diagrams were designed with BioRender (2023), and all figures were finalised and curated using Adobe Illustrator (AI) 2023.

## Statistics

Only biological replicates were used with no technical replicates. Data are presented as group means ± SD or medians ± IQR with individual participants/animals shown as symbols. All data were analysed using GraphPad Prism version 9 and R Studio version 4.4.1. Normal distribution of samples was tested using the D'Agostino & Pearson test. For experiments containing only two independent groups, normally distributed data were analysed using a two-tailed unpaired t-test, while when data did not pass normality tests, a two-tailed Mann-Whitney test was used to determine differences between groups. For experiments that included more than two groups and were non-normally distributed, Kruskal-Wallis tests were used, followed by Dunn's post-hoc test for pairwise comparisons. Lines on summary dot plots represent mean ± SD for normally distributed groups and median ± IQR for non-normal groups. To account for multiple testing, the significance threshold of <0.05 was adjusted by dividing it by the total number of comparisons being made using Bonferroni's correction for multiple testing unless otherwise stated. Multiple linear regression analysis was performed using the R function lm, and only cases with no missing data were included in the analysis. Least absolute shrinkage and selection operator (LASSO) regression analysis was performed using the R package glmnet.

## Reporting summary

Further information on research design is available in the Nature Portfolio Reporting Summary linked to this article.

## Data availability

The RNA sequencing data have been deposited in EGA under study accession EGAS50000001123 and dataset accession EGAD50000001616 (www.ega-archive.org/studies/EGAS50000001123). Data access requests should be submitted via the CLUSTER consortium data access committee (https://www.clusterconsortium.org.uk/researchers/cluster-datasets-and-data-access/). Please find the CLUSTER consortium data access schematic in Supplementary data file 2. All data that are included in this manuscript are in the supplementary materials or available from the authors, as are the unique reagents used in this article. The raw numbers for charts and graphs are available in the Source Data file wherever possible. Source data are provided in this paper.

## Code availability

R analysis was performed using packages detailed in the Methods. However, for B-cell receptor somatic hypermutation analysis, the analysis code can be found in the supplementary data file 1.

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

## Acknowledgements

We thank all families, patients, parents and carers who contributed samples and data for this work. We thank the UCL Genomics facility for sequencing (P. Niola and T. Brooks) and the UCL Flow Cytometry Facility for cell sorting (A. Eddaoudi, M. Palor and P. Constandinou) and general flow cytometry assistance (Jamie Evans). We thank the clinical study team for support in recruitment and data collection, including K. Kupiec, S. Cruickshank-Hull and the CHARMS Study group. We are grateful to the CLUSTER champions and members of the CLUSTER consortium for invaluable input and feedback. We thank the Childhood Uveitis Study Steering Patient Involvement Group for their support. We thank Professor A. Dick for ophthalmology expertise and advice on patient stratification, and Professor S. Coupland for ophthalmic histopathology expertise and oversight for plasma cell identification. B.J. was funded by a Versus Arthritis Fight 4 Sight PhD studentship (U/24VA22; to L.R.W. and E.C.R.). B.I., V.A. and E.C.R. are funded by a Kennedy Trust for Rheumatology Research Senior Research Fellowship (KENN 21 22 09; to E.C.R.). P.J. wass supported by a FOREUM research career grant (094; to E.C.R.). This work is also supported by a Research Prize from the Lister Institute for Preventive Medicine (to E.C.R) and a Child and Adolescent Eye Health research grant from the Medical Research Foundation and Moorfields Eye Charity (MEC: CAEH–24-102; MRF: MRF-JF-EH-23-118; to E.C.R). C.J.C. is a Wellcome Clinical Research Career Development Fellow (224586/Z/21/Z). A.L.S. is supported by an NIHR Clinician Scientist award (CS-2018-18-ST2-005) and the Wellcome Trust (311252/Z/24/Z). L.R.W. and CLUSTER are supported by the Medical Research Council (MR/R013926/1), Versus Arthritis (22084), Great Ormond Street Hospital Children's Charity (VS0518), Olivia's Vision, and NIHR GOSH BRC (BRC-1215-20012). L.R.W. was also supported by Versus Arthritis Centre for Excellence grant (21593) and by an NIHR Senior Investigator award. C.W was additionally supported by Wellcome Trust (WT220788) and the MRC (MC UU 00002/4). WYL was supported by the NIHR Cambridge BRC (BRC-1215-20014). The views expressed are those of the authors and not necessarily those of the NHS, NIHR or the Department of Health.

## Author contributions

B.J. designed experiments, performed experiments, analysed data and co-wrote the manuscript. B.I. and W-Y.L. performed BCR repertoire analysis. B.I. performed SHM analysis. V.A. and P.J. performed experiments and analysed data. M.K. curated data. R.R. and W-Y.L. performed RNA-seq QC. R.R. performed all analyses included in this manuscript at UCL however he is now based at Imperial College London (email: r.restuadi@lms.mrc.ac.uk). C.W. provided statistical expertise and critically reviewed the manuscript. J.K. performed H&E staining and analysis; Y.M. assisted with tissue preparation. C.J.C. provided expertise in experimental autoimmune uveitis, access to ocular samples and critically reviewed the manuscript. A.L.S. provided access to ocular samples, clinical expertise on uveitis and critically reviewed the manuscript. L.R.W. obtained funding, provided clinical expertise on JIA, is the CI of the CLUSTER Consortium and critically reviewed the manuscript. E.C.R. conceptualised the study, designed and performed experiments, obtained funding, supervised the study and co-wrote the manuscript.

## Competing interests

L.R.W. declares consultancies with Pfizer and Cabaletta unrelated to this work and research funding from Pfizer Inc. for a separate project. C.W. receives funding from MSD and GSK and is a part-time employee of GSK. The CLUSTER Consortium has received support through contributions-in-kind from GSK, Pfizer and UCB, and research funding from AbbVie Inc., Lilly and SOBI. These organisations did not contribute to the planning or analysis of this work. All other authors declare no competing interests.

## Additional information

## the CLUSTER consortium

Lucy R. Wedderburn[2,8,9], Zoe Wanstall[2], Vicky Alexiou[1,3], Fatjon Dekaj[2], Bethany R. Jebson[1,2,3], Melissa Kartawinata[1,2], Aline Kimonyo[2], Eileen Hahn[2], Genevieve Gottschalk[2], Freya Luling Feilding[2], Alyssia McNeece[2], Fatema Merali[2], Elizabeth Ralph[2,8], Emily Robinson[2], Emma Sumner[2], Andrew Dick[4,7], Michael W. Beresford[10], Emil Carlsson[10], Joanna Fairlie[10], Jenna F. Gritzfeld[10], Oliver McClurg[10], Karen Rafferty[10], Athimalaipet V. Ramanan[11], Teresa Duerr[11], Michael Barnes[12], Sandra Ng[12], Kimme Hyrich[13], Stephen Eyre[13], Soumya Raychaudhuri[13], Wendy Thomson[13], John Bowes[13], Jeronee Jennycloss[13], Saskia Lawson-Tovey[13], Paul Martin[13], Andrew Morris[13], Stephanie Shoop-Worrall[13], Samantha Smith[13], Michael Stadler[13], Damian Tarasek[13], Melissa Tordoff[13], Annie Yarwood[13], Chris Wallace[5,6], Wei-Yu Lin[5,6], Prof Nophar Geifman[14] & Sarah Clarke[15]

[10]University of Liverpool, Liverpool, UK. [11]University Hospitals Bristol and Weston NHS Foundation Trust, Bristol, UK. [12]Queen Mary University of London, London, UK. [13]University of Manchester, Manchester, UK. [14]University of Surrey, Guildford, UK. [15]School of Population Health Sciences and MRC Integrative Epidemiology Unit, University of Bristol, Bristol, UK.

