## [Transparent Peer Review file · Nature Communications]

Altered B cell activation contributes to the immunopathogenesis of childhood arthritis-associated uveitis

Corresponding Author: Dr Elizabeth Rosser

Version 0:

Reviewer comments:

Reviewer #1

(Remarks to the Author)

This is a well-conceived, comprehensive study that addresses a significant gap in our understanding of the immunopathogenesis of juvenile idiopathic arthritis-associated uveitis (JIA-uveitis). The work provides compelling evidence for a pathogenic role of heightened germinal center (GC) reactions and DN1 B-cell expansion in JIA-uveitis, challenging the prevailing notion of a predominantly T cell-driven disease. The inclusion of both human and animal model data strengthens the translational impact of the findings.

Major Comments

1. The authors should more clearly articulate how their findings compare to recent reports in other pediatric autoimmune diseases (e.g., SLE, IgA nephropathy) where DN1 B cells and GC dysregulation have been implicated. A comparative discussion will enhance the translational relevance.

2. Patient Cohort Heterogeneity:

The clinical heterogeneity (JIA subtypes, treatments, stage of disease) is acknowledged. Please expand on how these variables were controlled statistically, and discuss potential confounding effects more explicitly.

3. Gene Expression Data Interpretation:

While bulk RNA-seq of CD19+ B-cells showed limited differential gene expression between groups, it is possible that signals from rare, disease-driving subsets (e.g., DN1 cells) were masked. Single-cell RNA-seq or subset-specific transcriptomics would further validate nuances in B-cell activity.

4. Clinical Metrics and Outcome Correlation:

Are DN1 B-cell frequency and BCR clonality correlated with clinical severity, progression, or treatment response in JIA-uveitis? Such data would strengthen the case for clinical utility.

5. Animal Model Limitations:

As acknowledged by the authors, the EAU model does not fully recapitulate human JIA-uveitis (joint disease is absent, uveitis type differs). Discuss how these differences may affect translational projection and therapeutic expectations.

6. Future Directions:

Discuss the feasibility and potential timeline for clinical translation of CD40L antagonists in pediatric uveitis. Are there safety concerns or ongoing trials in related conditions that may expedite development?

Minor Comments

- The sample size for AqH and enucleated tissues is understandably limited, but more information about patient clinical characteristics (disease duration, therapy, ANA status) should be provided in supplementary tables.
- Certain figures (e.g., flow cytometry plots, gating strategies) could benefit from additional annotation for improved clarity.
- Ensure all references, especially those relating to novel B-cell subsets, are up-to-date and provide context for readers outside the field.
- Summary tables comparing patient demographics and major findings may aid reader comprehension.
- At times, DN1 B cells are referred to as “recent GC emigrants” and elsewhere as “potential long-lived memory.” The manuscript should consistently clarify the developmental interpretation adopted by the authors.

Reviewer #2

(Remarks to the Author)

The authors have explored the role of B cells in uveitis seen in patients with JIA. They demonstrate an increase in DN B cell in children with uveitis, which had good correlation with memory B cells and suggest early GC immigrants. In 3 eye tissue got at the time of enucleation, they showed infiltration by plasma cell however the B cell infiltration is not clear. Though single cell RNA seq of B cells did not show much difference in clonality between those with and without uveitis they found some minor differences

Further in IRBP induced uveitis model the authors showed B cell infiltration which is reduced on treatment with CD40L antagonist and as well as in bc6 KO mice that has less of GC reaction as GC reaction is probably the source of DN B cells. Based on these, The authors speculate if CD40 blockade may be useful in uveitis.

It is an interesting piece of data but due to samples having been collected long ago for another study the population is very heterogeneous. An ideal study should have included prospectively enrolled oligo JIA patients (all ANA positive) with and without uveitis at an early stage of disease so that the drugs do not impact the immune cells. Follow up of those patients would have provided if the number of DN1 cells comes down on treatment or not.

It is surprising that no significant difference in BCR was found between the two groups. How many CD19+ B cells sequenced? Could this be due to many different types of JIA as each has a different immunopathogenesis. Further the distribution of most patient variables are different between those with and without uveitis as each of this variable can affect immune cells. There was a stark gender difference in 2 groups in children especially near puberty the immune cells numbers etc can be different between the boys and girls. Can the authors do a sensitivity analysis and compare the oligo JIA patients who form the majority in both groups (with and without uveitis) to see if that shows the same findings in B cell or not?

The average duration of disease is 4 years but what is the duration of uveitis? Significant number of patients were on steroids and or TNF therapy?

The animal model used is not an ideal model as it is a T cell mediated model while the authors want to show the role of B cells. Could another model using organoids given better results?

However despite these caveats the paper shows some involvement of B cells in uveitis of JIA. Use of animal model though not perfect adds credence to their hypothesis.

Reviewer #3

(Remarks to the Author)

The authors investigate B cell populations and phenotypes in JIA-associated uveitis and conclude that B cells play a pronounced role in uveitis pathophysiology. This is an elegant study with important findings.

Main concerns of this referee are linked to conclusions and clarity of data presentation/discussion.

The conclusion that antibody positivity argues for a role of B cells in uveitis is maybe slightly strong. Rheumatoid factor positive patients, for instance, respond to rituximab but are 'protected' from uveitis (while then more severe, RF positive patients less frequently experience uveitis). This should be considered and mentioned/discussed in the manuscript.

In this referee's opinion the findings are important, indeed, exciting and definitely worth reporting and following but do not deliver proof of the 'upstream' involvement of B cells in disease pathology. B cell activation and proliferation may still be the result of pathological T cell activation and function. Proof (and even this would be 'incomplete' because of the complex interactions of immune cells) could be added by depleting B cells in uveitis, which is currently not commonly used in JIA-associated uveitis.

However, the authors propose that B cells play a key role as an early disease mechanism. They reduce inflammation in a mouse model by blocking CD40L. As CD40L also activated CD8+ T cells, this conclusion may be challenged. Would it be helpful to include B cell depletion through targeting of B cells specific surface antigens in this model?

Table 1: The percentage of HLA-B27 positive participants among uveitis patients appears very low (general population: 8-10%). Does this mean that 2% were positive among all tested participants or among all while not tested were counted as negative? While this likely is not meaningful for any conclusions here, the numbers/proportions appear confusing. This referee does not understand how 5% of patients can be classed as RF+ JIA but 9% were RF positive in the wider cohort? Were some incorrectly classified? Were some patients only tested once and RF were low titer and maybe false positive (after an infection, etc.)?

Nomenclature is somewhat outdated. The authors may want to talk about 'ancestry' (not ethnicity) and use the term "European ancestry" instead of 'Caucasian'.

Taken together, this is an interesting paper with some beautiful data. However, there should be more focus on alternative explanations, including T cell pathology. From the beginning, the authors could be more transparent about the over-representation of ANA positive oligo JIA patients in their cohort. There may well be differential pathomechanisms involved across the JIA subforms. This could be reflected in the title even.

Version 1:

Reviewer comments:

Reviewer #1

(Remarks to the Author)

This revised manuscript by Jebson et al. represents a significant advance in understanding the immunopathogenesis of JIA-associated uveitis. The authors have thoroughly addressed all reviewer comments with additional analyses, clearer discussion, and improved data presentation.

Reviewer #2

(Remarks to the Author)

The authors has now presented a much revised manuscript and all my queries have been addressed satisfactorily

Reviewer #3

(Remarks to the Author)

The authors have sufficiently addressed my comments and concerns.

Reviewer's Comments – Jebson et al., 2025, NCOMMS-25-45139

We would like to thank the reviewers for their insightful comments and suggestions on our manuscript. We were pleased they thought our manuscript was 'well-conceived' 'interesting' and 'elegant' and 'addresses a significant gap in the understanding of the immunopathogenesis of JIA-Uveitis'. Based on their comments, we have carefully reworded the text, added a more in-depth discussion of certain points and added new data. Please see below for our point-by-point response to individual queries below in blue and changes to the manuscript highlighted in grey.

Reviewer #1 (Remarks to the Author)

This is a well-conceived, comprehensive study that addresses a significant gap in our understanding of the immunopathogenesis of juvenile idiopathic arthritis-associated uveitis (JIA-uveitis). The work provides compelling evidence for a pathogenic role of heightened germinal center (GC) reactions and DN1 B-cell expansion in JIA-uveitis, challenging the prevailing notion of a predominantly T cell-driven disease. The inclusion of both human and animal model data strengthens the translational impact of the findings.

Major Comments (1 - 6)

1. The authors should more clearly articulate how their findings compare to recent reports in other paediatric autoimmune diseases (e.g., SLE, IgA nephropathy) where DN1 B cells and GC dysregulation have been implicated. A comparative discussion will enhance the translational relevance.

We would like to thank the reviewer for this important suggestion. We have now provided additional information regarding how our findings relate to previous findings that DN1 B cells are expanded in primary anti-phospholipid syndrome (PAPS)¹ and IgA nephropathy² (see lines 429-446).

2. Patient Cohort Heterogeneity. The clinical heterogeneity (JIA subtypes, treatments, stage of disease) is acknowledged. Please expand on how these variables were controlled statistically and discuss potential confounding effects more explicitly.

The reviewer correctly identifies the substantial clinical heterogeneity present in our cohorts which reflects the complex nature of JIA as a disease. We have addressed this heterogeneity through multiple statistical approaches: 1) **Multiple Linear Regression Analysis (Figure 1.K)** where CD11c- (DN1) B cells were inputted as the dependent variable and key clinical characteristics (age, sex, ANA status, 'polygo' subtype, and treatment) as independent variables. This analysis demonstrated that uveitis status had the most significant effect on changes in CD11c- DN1 B cells, even after controlling for these confounders; 2) **Penalised LASSO Regression (Figure 3.B)** incorporating all available clinical demographic information (treatments, sex, age etc.) alongside 34 different immune cell populations. This unbiased approach identified CD11c- DN B cells, memory B cells, reduced Th2 T cells, positive ANA status and oligoarticular JIA subtype as the most important factors associated with uveitis presence and excluded 43 other clinical variables and immune cell proportions; 3. **Adjustment in Bulk RNAseq Analysis of the B cell receptor repertoire and**

differentially expressed genes (Figure 3. F-G and Supplementary Figure 6) for age and sex. We have also now added new data which includes further subgroup analysis of JIA patients with oligoarticular arthritis only and have demonstrated the same effect in DN1 B cell expansion in JIA patients with uveitis patients (**Figure 1 I-J**). As suggested by the reviewer we have now extended our discussion of these confounders and how we have adjusted for them in our different analyses more explicitly (see lines 347-360 and 381-385).

3. Gene Expression Data Interpretation. While bulk RNA-seq of CD19+ B-cells showed limited differential gene expression between groups, it is possible that signals from rare, disease-driving subsets (e.g., DN1 cells) were masked. Single-cell RNA-seq or subset-specific transcriptomics would further validate nuances in B-cell activity.

We fully agree with the reviewer that single-cell RNA sequencing (scRNAseq) would be extremely valuable for unmasking transcriptomic differences in rare B-cell subsets, particularly DN1 cells, that may be diluted in bulk RNAseq analysis. However, logistical constraints have prevented this approach in our current study. The paediatric blood samples analysed in this study were collected over a 20+ year period as part of a collaborative consortium effort (www.clusterconsortium.org.uk) and were originally analysed between 2019-2023 by flow cytometry and bulk RNA sequencing of CD4+ T cells, CD8+ T cells, CD14+ monocytes and CD19+ B cells to understand how to predict treatment response in JIA based on arthritic (joint) inflammation). We have now utilised this rich dataset retrospectively and re-analysed the data to explore the immunopathogenesis of JIA-uveitis. Unfortunately, this means that there are no remaining blood samples from this current cohort, and replenishment of the biobank for new studies has been severely impacted by the COVID-19 pandemic, which limits our ability to perform subset-specific transcriptomics or scRNAseq at this time. Over the course of the next 2-3 years, we are aiming to recruit JIA-uveitis patients as part of our newly funded uveitis-prospective study (created in response to the data gathered in this study) and JIA patients with joint involvement alone as part of ongoing bio-sampling of this population to perform this experiment. Unfortunately, due to the relative rarity of JIA-uveitis, we cannot collect these samples to perform a powered analysis any quicker. We hope that this reviewer agrees that timely publication of our current findings preclude the inclusion of further transcriptomic data in this manuscript and have now addressed how these data would strengthen future studies in two sections in the discussion (see lines 404-407 and 443-446). In addition, after reflection regarding how we could use our current data to strengthen our findings and to further validate differences in cell activity, we have performed additional analysis of the BCR repertoire to assess the levels of somatic hypermutation (SHM) alongside clonality. These new data demonstrate that the BCR of JIA-uveitis patients is significantly more mutated than that of JIA patients with arthritis alone ($p=0.0005$). As SHM takes place uniquely within the germinal centre, these new data provide further evidence that GC responses are dysregulated in JIA-uveitis (**Figure 3.G**).

4. Clinical Metrics and Outcome Correlation: Are DN1 B-cell frequency and BCR clonality correlated with clinical severity, progression, or treatment response in JIA-uveitis? Such data would strengthen the case for clinical utility.

Many thanks for this suggestion. Unfortunately, these samples were originally collected for JIA-arthritis studies through rheumatology clinics, which only recorded

uveitis status as previous/current/no without additional ophthalmological details such as disease severity or uveitis-specific treatment responses. While treatment response data exists for the JIA arthritis patients in this study³, this means that no equivalent data is available for the uveitis as patients were recruited to this study based on joint rather than eye inflammation at time of sampling. This limitation is now discussed further in the manuscript (see lines 562-570). Interestingly, stratification based on available data (whether patients had active or inactive uveitis at time of sample) demonstrates that DN1 B cell frequency remains high in patients with inactive disease (of which the majority are on systemic medications) suggesting that current medications do not impact this underlying defect (see **Figure 1G-H and Supplementary Table 2**). We have also revisited a subset of patients excluded from our original analysis due to their small sample size, who went on to develop uveitis after sample date (n=10). These data demonstrate that the increase in DN1 B cell does not precede uveitis development, these data are included below for the reviewer's interest (**Figure R1**).

Figure R1. Dot plot showing the proportion of DN and CD11c- DN B cells CD19+ B cells in the peripheral blood (PB) of JIA patients with uveitis (n=44), and patients who went on to develop uveitis after the time of sample - future uveitis (n=10). Lines in dot plots represent median \pm IQR. Kruskal-wallis test was used to determine significance of difference between groups.

5. Animal Model Limitations: As acknowledged by the authors, the EAU model does not fully recapitulate human JIA-uveitis (joint disease is absent, uveitis type differs). Discuss how these differences may affect translational projection and therapeutic expectations.

We acknowledge the differences between EAU and human JIA-uveitis will affect translational interpretation and this is now expanded on in the discussion (see lines 522-549). We appreciate that results should be interpreted as proof-of-concept that targeting dysregulated GC reactions to treat JIA-uveitis could be a future therapeutic avenue. However, it is important to note that the EAU model has previously provided valuable mechanistic evidence that has informed therapeutic design for human uveitis including the use of cyclosporin, TNF α and IFN α ⁴⁻⁷ underlying the potential important 'first step' in the path to translation our data represents. We have now highlighted this

in the text and provided new examples that would strengthen translation potential of our findings including organoids systems as part of future studies (see lines 581-592).

6. Future Directions: Discuss the feasibility and potential timeline for clinical translation of CD40L antagonists in paediatric uveitis. Are there safety concerns or ongoing trials in related conditions that may expedite development?

Early CD40L antagonists were associated with serious thromboembolic complications due to platelet binding and activation⁸. However, next-generation CD40L inhibitors have been specifically engineered with reduced platelet binding affinity, substantially improving their safety profile and resulting in their trial in various adult autoimmune conditions. There are 2 phase 3 trials underway in multiple sclerosis (MS): using Frexalimab to treat relapsing remitting MS in adults (trial numbers: NCT06141473 and NCT06141486). In sjogrens, a phase 2 Dazodalibep trial showed promising results⁹. and in SLE and type I Diabetes, Frexalimab is currently being trialled¹⁰. We have added a line to the discussion to highlight how these agents (as they have passed safety testing) could be applied to JIA-uveitis (lines 542-549).

Minor Comments

- The sample size for AqH and enucleated tissues is understandably limited, but more information about patient clinical characteristics (disease duration, therapy, ANA status) should be provided in supplementary tables.

We agree with the reviewer regarding the importance of providing detailed patient clinical characteristics. We have detailed demographic tables for both aqueous humour and enucleated eye samples which are in the supplementary material (**Supplementary Table 4-5**). We have added disease duration where possible. For the enucleated tissues, we had obtained histopathological records which were usually linked to patient clinical records. However, as these samples were collected historically as part of routine clinical care (prior to the year 2000), minimal demographic or clinical information was recorded electronically. Furthermore, all three patient therapies were likely carried out in various private institutions, further complicating access to clinical records. Due to data protection laws and anonymisation, these records are unfortunately inaccessible. This represents a limitation inherent to archived tissue studies and is now noted in the text (see line 562-570).

- Certain figures (e.g., flow cytometry plots, gating strategies) could benefit from additional annotation for improved clarity.

We have added additional annotation to population labels within flow cytometry figures and included a detailed gating strategy including markers and associated fluorochromes to the supplementary figures for clarity (**Supplementary Figures 7-10**).

- Ensure all references, especially those relating to novel B-cell subsets, are up-to-date and provide context for readers outside the field.

Many thanks for this suggestion, we have updated the manuscript accordingly.

- Summary tables comparing patient demographics and major findings may aid reader comprehension.

A summary table comparing patients demographics can be found in **Table 1** (Blood Patient demographics), **Supplementary Table 2** (JIA-uveitis patient demographics active/inactive), **Supplementary Table 3** (JIA and JIA-uveitis patients with oligoarticular arthritis demographics), **Supplementary Table 4** (Aqueous humour patient demographics), **Supplementary Table 5** (Enucleated eye patient demographics). We have also included a graphical abstract in our new version of the manuscript to aid reader comprehension (**Figure 7**).

- At times, DN1 B cells are referred to as “recent GC emigrants” and elsewhere as “potential long-lived memory.” The manuscript should consistently clarify the developmental interpretation adopted by the authors.

We apologise for the inconsistent terminology regarding DN1 B cells and their developmental status. We have revised the manuscript to address this issue. These terms are now only quoted in the introduction where we introduce the literature of DN1 B cells and the different terminology’s used by others in the field (lines 88-91 and 215).

Reviewer #2 (Remarks to the Author) (1 - 5)

1. The authors have explored the role of B cells in uveitis seen in patients with JIA. They demonstrate and increase in DN B cell in children with uveitis, which had good correlation with memory B cells and suggest early GC immigrants. In 3 eye tissue got at the time of enucleation, they showed infiltration by plasma cell however the B cell infiltration is not clear. Though single cell RNA seq of B cells did not show much difference in clonality between those with and without uveitis they found some minor differences. Further in IRBP induced uveitis model the authors showed B cell infiltration which is reduced on treatment with CD40L antagonist and as well as in bc6 KO mice that has less of GC reaction as GC reaction is probably the source of DN B cells. Based on these, the authors speculate if CD40 blockade may be useful in uveitis. It is an interesting piece of data but due to samples having been collected long ago for another study the population is very heterogeneous. An ideal study should have included prospectively enrolled oligo JIA patients (all ANA positive) with and without uveitis at an early stage of disease so that the drugs do not impact the immune cells. Follow up of those patients would have provided if the number of DN1 cells comes down on treatment or not.

We thank the reviewer for their thoughtful summary of our findings and constructive suggestions regarding study design. We appreciate their recognition that sample heterogeneity reflects the challenges inherent in studying rare diseases over extended collection periods. We agree that the heterogeneous nature of our cohorts, collected over 20+ years as part of broader JIA studies represents a limitation. The reviewer correctly identifies that prospective enrolment with standardised protocols would be ideal for future investigations. Based on our data in this manuscript, we have recently established a uveitis-specific prospective cohort study that will record comprehensive ophthalmological data over time, allowing us to address many of the limitations

identified and will provide a rich resource for future studies. These studies will enable longitudinal tracking of DN1 B-cell frequencies in relation to treatment responses and disease progression, which as this reviewer correctly identifies is a fundamental next step. We have added a line in the discussion to address this important point (see lines 562-570).

As requested by this reviewer below (point 3), we have now performed subgroup analysis comparing DN and DN1 B cell frequencies in oligoarticular JIA patients only (n=45 JIA-alone vs n=38 JIA-uveitis). This analysis is presented in **Figure 11-J** and shows the same findings as using the full cohort that DN ($p=0.0161$) and DN1 ($p=0.0083$) B cells are increased in patients with uveitis. These data provide further evidence that the increase in DN and DN1 B cells is driven by uveitis status rather than differences in subtype between group. While we understand the reviewer's suggestion to focus exclusively on oligoarticular ANA-positive patients, we maintain a broader approach based on emerging evidence that ILAR clinical subtypes do not completely represent underlying "molecular pathotypes"¹¹. Our data supports this concept, showing that immunological signatures may not fully align with traditional ILAR classifications, and uveitis (while more common in ANA+ oligoarticular patients) can also develop in ANA- patients and patients with other subtypes of JIA. In further support of this concept, our recent study focusing on the immune cell infiltrate in JIA synovial tissue biopsies which demonstrates that age-of-onset or severity seem to have a greater impact on synovial tissue immunobiology in JIA rather than ILAR subtype and that JIA synovial tissue pathology is more homogenous than adult arthritides¹². Restricting studies to isolated ILAR categories may limit our understanding of shared pathogenic mechanisms that lead to uveitis regardless of subtype and exclude patients from clinical trials who could also benefit from new therapies.

2. It is surprising that no significant difference in BCR was found between the two groups. How many CD19+ B cells sequenced? Could this be due to many different types of JIA as each has a different immunopathogenesis.

Apologies for the confusion regarding our BCR data. To clarify, although we found no significant differences in differentially expressed genes between CD19+ B-cells from JIA-alone and JIA-uveitis patients (**Supplementary Figure 6**), we found significant differences in the clonal diversity of the B cell receptor (BCR) repertoire. Since our initial submission, we have also added new data demonstrating that the BCR repertoire in JIA-uveitis is significantly more somatically hypermutated compared to JIA-alone patients (**Figure 3G**). These data demonstrate that changes to the BCR repertoire are a more faithful representation of the history of B cells antigen-activation dynamics due to the transient nature of transcriptional regulation. Of note, we have also performed additional analysis comparing DEG between oligo patients alone and find no significant differences between groups (*data not shown*). We sequenced an average of ~300,000 B cells per patient (ranging from 50,000-500,000), this information has now been added to the methodology section (lines 838-840).

3. Further the distribution of most patient variables are different between those with and without uveitis as each of this variable can affect immune cells. There was a stark gender difference in 2 groups in children especially near puberty the immune cells numbers etc can be different between the boys and girls. Can the authors do a

sensitivity analysis and compare the oligo JIA patients who form the majority in both groups (with and without uveitis) to see if that shows the same findings in B cell or not?

We appreciate the reviewer highlighting sex differences between groups. This distribution reflects the well-established epidemiology of JIA-associated uveitis, where females have a 2:1 higher risk of developing uveitis compared to males¹³. However, regarding the reviewer's concern about pubertal effects on immune cell populations, we have recently published that B cell subsets profiles in peripheral blood remain similar in children regardless of sex until puberty¹⁴. Given that our cohort consists primarily of pre-adolescent¹⁵ (the majority of which will be pre-pubertal) children (median age 8.7 for JIA-uveitis and 9.1 for JIA), sex-related immune differences should be minimal. This is supported by our multiple linear regression model which identifies no association between sex and DN1 B cell frequency, and all our transcriptional and BCR data is adjusted for both age and sex. For the data from the subgroup analysis, please see our reply above (reviewer 2, point 1).

4. The average duration of disease is 4 years but what is the duration of uveitis?

Unfortunately, as noted in our response to reviewer 1, these patients were originally recruited for JIA arthritis that was limited to basic disease status (active/inactive/no) from rheumatology notes and detailed ophthalmological data including uveitis duration was could not be collected. This limitation is now extensively discussed in the manuscript (see lines 562-570).

4. Significant number of patients were on steroids and or TNF therapy?

TNF α therapy use was minimal in our cohorts (5% of JIA-uveitis patients and 0% of JIA-alone patients). We acknowledge the higher prevalence of systemic steroid use: 32% of JIA-uveitis patients versus 4% of JIA-alone patients. To address potential confounding effects, we included systemic steroid treatment as a variable in both our multiple linear regression (**Figure 1.K**) and penalised LASSO regression (**Figure 3.B**) analyses, as detailed in our response to reviewer 1 (point 3). Both statistical models demonstrated that systemic steroids did not significantly contribute to CD11c⁻ DN1 B-cell increases and were not identified as an important factor affecting uveitis likelihood. These analyses confirm that our key findings regarding DN1 B-cell elevation are independent of steroid treatment effects. For clarity, we have standardised our terminology throughout the manuscript to ensure all references to systemic steroids use consistent language and apologise for any confusion in the original submission when oral steroid and systemic steroid were used interchangeably.

5. The animal model used is not an ideal model as it is a T cell mediated model while the authors want to show the role of B cells. Could another model using organoids given better results? However, despite these caveats the paper shows some involvement of B cells in uveitis of JIA. Use of animal model though not perfect adds credence to their hypothesis.

We acknowledge that the EAU model is traditionally viewed as T-cell mediated. However, we are not the first group to investigate B-cell contributions in this model. Previous studies have demonstrated important B cell roles in EAU pathogenesis¹⁶⁻¹⁸, and B/plasma cell-targeted therapies have shown therapeutic efficacy^{19,20}. We also

believe that because EAU is primarily driven by T cells, the observation of a strong B cell component provides more compelling evidence of B cell involvement than would be obtained from a model inherently designed around B cell activation. These findings support the relevance of studying B cell biology, and specifically for our study germinal centre reactions, in this established uveitis model. Regarding organoid models, uveal organoid systems, which are the most appropriate model for interrogating the anterior uveitic disease observed in JIA-uveitis patients, are currently not available with no standardised methodology, and there are no published studies for uveal organoids with immune cell co-cultures. We agree this would be an exciting future direction for the field and have highlighted this in the discussion (see lines 588-592).

Reviewer #3 (Remarks to the Author)

The authors investigate B cell populations and phenotypes in JIA-associated uveitis and conclude that B cells play a pronounced role in uveitis pathophysiology. This is an elegant study with important findings.

Main concerns of this referee are linked to conclusions and clarity of data presentation/discussion. The conclusion that antibody positivity argues for a role of B cells in uveitis is maybe slightly strong. Rheumatoid factor positive patients, for instance, respond to rituximab but are 'protected' from uveitis (while then more severe, RF positive patients less frequently experience uveitis). This should be considered and mentioned/discussed in the manuscript.

We thank the reviewer for highlighting this important observation regarding the complexity of the role of B cells and antibodies in uveitis. We have added a discussion of the RF+ poly uveitis paradox (lines 480-487) where we acknowledge that it is likely that different B cell subsets or activation states may have different roles in arthritis and uveitis.

In this referee's opinion the findings are important, indeed, exciting and definitely worth reporting and following but do not deliver proof of the 'upstream' involvement of B cells in disease pathology. B cell activation and proliferation may still be the result of pathological T cell activation and function. Proof (and even this would be 'incomplete' because of the complex interactions of immune cells) could be added by depleting B cells in uveitis, which is currently not commonly used in JIA-associated uveitis. However, the authors propose that B cells play a key role as an early disease mechanism. They reduce inflammation in a mouse model by blocking CD40L. As CD40L also activated CD8+ T cells, this conclusion may be challenged. Would it be helpful to include B cell depletion through targeting of B cells specific surface antigens in this model?

We thank the reviewer for these important observations, which for clarity we have answered together. We appreciate the wide range of effects of CD40L, which are now more broadly discussed (lines 534-537). These broad ranging effects are exactly why we complemented these experiments with data demonstrating that CD4^{cre}BCL6^{fl/fl} mice develop significantly less severe disease than EAU controls (see **Figure F-J**). CD4^{cre}BCL6^{fl/fl} mice are well published to serve as a T follicular helper cell knockout mouse (leaving the rest of the T cell compartment 'intact') and cannot mount GC reactions. Thus, we used this system as a model to specifically investigate the role of

the germinal centre in the pathology of EAU based on our human findings further to the broad impact of anti-CD40L. As rightly pointed out by this reviewer, these experiments provide evidence that changes to T cell:B cell interaction within the germinal centre influence EAU pathology and potentially human uveitis. Throughout the manuscript we have rewritten the text to provide a more balanced perspective focusing on the communication between cell types, how this can drive uveitis pathology and the role the B cell may take in this dynamic relationship. We have also added data to our mouse studies highlighting the impact of EAU and our interventions on T follicular helper cells (see **Figure 5 E-F, Figure 6 F-G and M-N**) to ensure a more balanced assessment of both B-T cells within the germinal centre. In addition, while researching the best methodology to perform the B cell specific depletion suggested by this reviewer, we have excitingly found a recently published report in *Nature Communications*¹⁹ that has performed this exact experiment demonstrating that depletion of B cell post-EAU onset, but not prior to onset, suppresses EAU severity (*paper figure 5 f-g*). These data support our pilot analyses that the increase in DN1 B cells does not precede disease development (see response to reviewer 1, point 4) and rather that B cells seem to be driving part of the active disease process. For the sake of novelty, we have not repeated this experiment but have made sure this reference is highlighted in the introduction (see lines 77-80) and have now extensively discussed how the findings from this previous paper complements our own alongside the translational relevance of both studies (see lines 500-503, 522-532). Our apologies for missing this important reference from our original submission.

Table 1: The percentage of HLA-B27 positive participants among uveitis patients appears very low (general population: 8-10%). Does this mean that 2% were positive among all tested participants or among all while not tested were counted as negative? While this likely is not meaningful for any conclusions here, the numbers/proportions appear confusing.

We thank the reviewer for this observation and have corrected a calculation error. In **Table 1** in our JIA-uveitis cohort, only 13 of 44 patients (30%) were tested for HLA-B27, with 1 testing positive (8%). In the JIA group, 53 of 116 patients (46%) were tested, with 16 testing positive (30%). The large amount of missing HLA-B27 in our cohort is because the HLA-B27 test is not used routinely in all JIA patients at the centre these samples were collected from. We acknowledge this is a limitation of our study and have discussed this in the methods (lines 716-718).

This referee does not understand how 5% of patients can be classed as RF+ JIA but 9% were RF positive in the wider cohort? Were some incorrectly classified? Were some patients only tested once and RF were low titer and maybe false positive (after an infection, etc.)?

We thank the reviewer for seeking clarification on this point and apologies for any confusion. We believe there may have been a misreading of Table 1. In the JIA-uveitis group, 7% had missing RF data, therefore 4 out of 37 tested patients were RF+ (10.8%, rounded to 11%). In the JIA group, 9% had missing RF data, so 11 out of 107 tested patients were RF+ (10.3%, rounded to 10%).

Nomenclature is somewhat outdated. The authors may want to talk about 'ancestry' (not ethnicity) and use the term "European ancestry" instead of 'Caucasian'.

We appreciate the reviewer's point regarding updated nomenclature for ancestry. While we agree that terms like 'European ancestry' are preferable to 'Caucasian,' we have maintained the original terminology as recorded in the source studies (dating from the late 1990s to 2019) to ensure data integrity and avoid potential misclassification. We have however changed the subheading from 'ethnicity' to 'ancestry' in our demographic tables. We now highlight this limitation in our methods section (lines 712-714).

Taken together, this is an interesting paper with some beautiful data. However, there should be more focus on alternative explanations, including T cell pathology. From the beginning, the authors could be more transparent about the over-representation of ANA positive oligo JIA patients in their cohort. There may well be differential pathomechanisms involved across the JIA subforms. This could be reflected in the title even.

We have added upfront acknowledgment of the oligoarticular ANA+ patient overrepresentation in our results section (lines 164-167), as this reflects the natural epidemiology of JIA-uveitis²¹ and as correctly noted by this reviewer, it is important to highlight this from the outset of our manuscript. We have also added discussion on the role of T cells/autoantibody producing cells throughout the manuscript and expanded on the idea that JIA-uveitis pathology may be driven by T cell:B cell interactions within the GC (lines 603-611).

References

1. Popova, A. *et al.* IgA class-switched CD27 – CD21 + B cells in IgA nephropathy. <https://doi.org/10.1093/ndt/gfae173> doi:10.1093/ndt/gfae173.
2. Dieudonné, Y. *et al.* Defective germinal center selection results in persistence of self-reactive B cells from the primary to the secondary repertoire in Primary Antiphospholipid Syndrome. *Nature Communications* 2024 15:1 **15**, 1–18 (2024).
3. Kartawinata, M. *et al.* Identification and validation of interferon-driven gene signature as a predictor of response to methotrexate in juvenile idiopathic arthritis. *Ann Rheum Dis* **84**, 1412–1424 (2025).
4. Khera, T. K. *et al.* Tumour necrosis factor-mediated macrophage activation in the target organ is critical for clinical manifestation of uveitis. *Clin Exp Immunol* **168**, 165 (2012).
5. Nussenblatt, R. B. BENCH TO BEDSIDE: NEW APPROACHES TO THE IMMUNOTHERAPY OF UVEITIC DISEASE. *Int Rev Immunol* **21**, 273–289 (2002).
6. Sharma, S. M., Nestel, A. R., Lee, R. W. J. & Dick, A. D. Clinical review: Anti-TNFalpha therapies in uveitis: perspective on 5 years of clinical experience. *Ocul Immunol Inflamm* **17**, 403–414 (2009).

7. Bansal, S., Barathi, V. A., Iwata, D. & Agrawal, R. Experimental autoimmune uveitis and other animal models of uveitis: An update. *Indian J Ophthalmol* **63**, 211 (2015).
8. Karnell, J. L., Rieder, S. A., Ettinger, R. & Kolbeck, R. Targeting the CD40-CD40L pathway in autoimmune diseases: Humoral immunity and beyond. *Adv Drug Deliv Rev* **141**, 92–103 (2019).
9. St. Clair, E. W. *et al.* CD40 ligand antagonist dazodalibep in Sjögren's disease: a randomized, double-blinded, placebo-controlled, phase 2 trial. *Nat Med* **30**, 1583–1592 (2024).
10. Fatima, T. *et al.* Frexalimab (SAR441344) as a potential multiautoimmune disorder tackling mAB targeting the CD40-CD40L pathway undergoing clinical trials: a review. *Annals of Medicine and Surgery* **86**, 7305 (2024).
11. Wedderburn, L. R., Ramanan, A. V & Croft, A. P. Towards molecular-pathology informed clinical trials in childhood arthritis to achieve precision medicine in juvenile idiopathic arthritis. *Ann Rheum Dis* **0**, 1–8 (2022).
12. Bolton, C. *et al.* Synovial tissue atlas in juvenile idiopathic arthritis reveals pathogenic niches associated with disease severity. *Sandrine Compeyrot-Lacassagne* **17**, 21 (2025).
13. Heiligenhaus, A., Minden, K., Föll, D. & Pleyer, U. Uveitis in Juvenile Idiopathic Arthritis. *Dtsch Arztebl Int* **112**, 92–100 (2015).
14. Peckham, H. *et al.* Estrogen influences class-switched memory B cell frequency only in humans with two X chromosomes. *J Exp Med* **222**, (2025).
15. Adolescent health. https://www.who.int/health-topics/adolescent-health#tab=tab_1.
16. Yu Jin Kyeong Choi Anita Uche Charles E Egwuagu, C.-R. N. & Charles Egwuagu, C. E. Production of IL-35 by Bregs is mediated through binding of BATF-IRF-4-IRF-8 complex to il12a and ebi3 promoter elements. *J Leukoc Biol* **104**, 1147–1157 (2018).
17. Choi, J. K. & Egwuagu, C. E. Analysis of regulatory B cells in experimental autoimmune uveitis. *Methods in Molecular Biology* **2270**, 437–450 (2021).
18. Oladipupo, F. O. *et al.* STAT3 deficiency in B cells exacerbates uveitis by promoting expansion of pathogenic lymphocytes and suppressing regulatory B cells (Bregs) and Tregs. *Scientific Reports* 2020 10:1 **10**, 1–14 (2020).
19. Li, H. *et al.* Multicellular immune dynamics implicate PIM1 as a potential therapeutic target for uveitis. *Nat Commun* **13**, (2022).
20. Neubert, K. *et al.* The proteasome inhibitor bortezomib depletes plasma cells and protects mice with lupus-like disease from nephritis. *Nature Medicine* 2008 14:7 **14**, 748–755 (2008).
21. Petty, R. E. & Zheng, Q. Uveitis in juvenile idiopathic arthritis. *World J Pediatr* **16**, 562–565 (2020).